# TFIIE orchestrates the recruitment of the TFIIH kinase module at promoter before release during transcription

Emmanuel Compe[1], Carlos M. Genes[1], Cathy Braun[1], Frederic Coin[1] & Jean-Marc Egly[1]

In eukaryotes, the general transcription factors TFIIE and TFIIH assemble at the transcription start site with RNA Polymerase II. However, the mechanism by which these transcription factors incorporate the preinitiation complex and coordinate their action during RNA polymerase II transcription remains elusive. Here we show that the TFIIEα and TFIIEβ subunits anchor the TFIIH kinase module (CAK) within the preinitiation complex. In addition, we show that while RNA polymerase II phosphorylation and DNA opening occur, CAK and TFIIEα are released from the promoter. This dissociation is impeded by either ATP-γS or CDK7 inhibitor THZ1, but still occurs when XPB activity is abrogated. Finally, we show that the Core-TFIIH and TFIIEβ are subsequently removed, while elongation factors such as DSIF are recruited. Remarkably, these early transcriptional events are affected by TFIIE and TFIIH mutations associated with the developmental disorder, trichothiodystrophy.

[1] Department of Functional Genomics and Cancer, IGBMC, CNRS/INSERM/University of Strasbourg, BP 163, 67404 Illkirch Cedex, C. U. Strasbourg, France. Correspondence and requests for materials should be addressed to E.C. (email: compe@igbmc.fr) or to J.-M.E. (email: egly@igbmc.fr)

In eukaryotes, accurate transcription of protein-coding genes requires the synchronized action of multiple molecular actors. Among them, the general transcription factors (GTFs), including TFIIA, TFIIB, TFIID, TFIIE, TFIIF and TFIIH assemble at the transcription start site with RNA Polymerase II (Pol II) to form the preinitiation complex (PIC). While the different GTFs have been characterized almost 30 years ago, little is known about the chronological order of the enzymatic activities that occurr during transcription initiation. Nonetheless, a conventional assembly order for a Pol II-PIC emerged, in which TBP (TATA Binding Protein), as part of TFIID, initially binds to the promoter and is further stabilized by TFIIB and TFIIA. Pol II and TFIIF then align to make a stable TFIID–TFIIA–TFIIB–Pol II–TFIIF promoter complex, which is finally completed by the addition of TFIIE and TFIIH[1,2].

The heterodimer TFIIE (composed of the TFIIEα and TFIIEβ subunits) seems to play a pivotal role in transcription by directly influencing the transition from initiation to elongation[3,4]. TFIIE interacts with different factors within the PIC, including Pol II[5,6] as well as with DNA immediately upstream of the transcription bubble region[7,8]. Furthermore, TFIIE seems to influence TFIIH activity[9], although it is not clear how this molecular process can occur.

TFIIH, which also intervenes during the nucleotide excision repair (NER) pathway, contains 10 subunits that can be resolved into two sub-complexes: the Core (composed of the subunits XPB, p62, p52, p44, p34, and TTDA) and the CAK (containing the CDK7 kinase, MAT1, and Cyclin H). Both sub-complexes interact with XPD subunit, whose ATPase/helicase activity promotes DNA opening during NER[10,11]. The role of TFIIH during transcription initiation mainly relies on its XPB and CDK7 enzymatic activities, XPD having essentially a structural function[12]. Observations revealed that XPB is not a conventional helicase[13,14], but an ATPase/translocase that pumps downstream DNA toward the Pol II to generate torsional and mechanical strain, leading to the formation of the DNA bubble[15–18]. For its part, CDK7 phosphorylates the serine 5 (Ser5-P) of the C-terminal domain (CTD) of the largest Pol II subunit (RPB1), thus contributing to promoter escape[19–21]. The CDK7 action is not restricted to Pol II, since other transcription factors are targeted by this kinase, including nuclear receptors (NRs), such as the retinoic acid receptor alpha (RARα)[22].

A better understanding of the transcription mechanism arose from the study of diseases related to mutations in genes encoding transcription machinery components (e.g. TFIIH, Mediator, etc.)[23,24]. Remarkably, mutations within the TFIIEβ subunit recently have recently been associated with trichothiodystrophy (TTD), an autosomal recessive developmental disorder mainly related to TFIIH mutations (mostly in *ERCC2/XPD* gene and few cases in *ERCC3/XPB* or *GFT2H5/p8/TTD-A* genes) and characterized by brittle hairs with alternating dark and light ("Tiger tail") banding with polarized microscopy and low content of sulfur-containing amino acids[25,26]. TFIIE/TTD patients, like TFIIH/TTD patients also have dry, ichthyotic skin, short stature, microcephaly, cerebellar dysfunction, developmental delay, happy engaging personality, attention deficit hyperactivity disorder, and myopia[27].

Our work reveals an unexpected dynamic process during which TFIIEα and TFIIEβ act as key factors to recruit the CAK of TFIIH within the PIC. Furthermore, we show that Pol II phosphorylation is accompanied by the release of the CAK and TFIIEα from the promoter, a process that takes place before DNA opening. Once the CAK and TFIIEα are released, RNA synthesis is initiated, a process during which the Core-TFIIH and TFIIEβ are also removed while elongation factors including DRB sensitivity-inducing factor (DSIF) are recruited. Strikingly, TTD-related mutations in either *XPD* or the TFIIEβ-coding gene

(*GTF2E2*) similarly disrupt these processes, which could explain why alterations of TFIIE or TFIIH lead to the same clinical syndrome.

## Results

**Defective TFIIE alters PIC formation.** Knowing that TTD cells with mutations in either XPB, XPD, or p8/TTD-A exhibit a decrease in the cellular concentration of TFIIH[28,29], we first examined the levels of components of the transcription machinery in fibroblasts isolated from TTD patients bearing either TFIIEβ/A150P or /D187Y mutations. Western Blot analysis showed that both α and β subunits of TFIIE were reduced in TTD/TFIIE-mutated cells when compared to wild-type fibroblasts (Fig. 1a). Such deficiency was only observed for TFIIE, since no reduction of the cellular concentration of TFIIH (visualized by XPB, XPD, and CDK7) or for the other GTFs, namely TFIIA (p35), TFIIB (p33), TFIID (TAF1, TAF4, and TBP), TFIIF (RAP74 and RAP30), and Pol II (visualized by the hypo-IIA and hyper-IIO phosphorylated form of its RPB1 subunit) was observed. We also evaluated the levels of TFIIE in TTD fibroblasts bearing the most common XPD mutations (Supplementary Fig. 1a). Surprisingly, we repeatedly observed a drop in TFIIE concentration in XPD/R722W fibroblasts and to a lower extend in XPD/R112H, which accompanied the decrease of TFIIH (Supplementary Fig. 1b).

We next investigated whether the expression of protein-coding genes was affected by TFIIE mutations, using the all-trans retinoic acid (t-RA) inducible *RARβ2* gene as a model. After t-RA treatment, the pattern of RARβ2 mRNA synthesis was reduced in both IIEβ/A150P and IIEβ/D187Y fibroblasts when compared with wild-type cells (Fig. 1b, c). Interestingly, similar profiles of RARβ2 deregulation were observed in XPD/R112H and /R722W fibroblasts (Supplementary Fig. 1c, d), revealing recurrent transcriptional failures in TTD.

Chromatin immunoprecipitation (ChIP) assays were then performed to determine whether deficiencies occurred during the PIC assembly. In wild-type fibroblasts, the recruitment of both Pol II (visualized by its RPB1 subunit, Fig. 1d, e) and TFIIH (CDK7, Fig. 1f, g) at the RARβ2 promoter highly increased at 6 h, then decreased 8 h post-treatment, which perfectly paralleled the profile of the RARβ2 mRNA synthesis (panels b and c). Conversely, in TFIIE-deficient fibroblasts, Pol II and TFIIH were barely recruited 6 h post-treatment, despite their unchanged cellular concentrations (Fig. 1a). Contrary to what was observed in normal fibroblasts, the Pol II and TFIIH recruitment tended to increase at 8 h in TFIIE-deficient fibroblasts. Defective recruitment of Pol II and TFIIH also occurred in XPD/R112H and XPD/R722W cells (Supplementary Fig. 1e, f, g, h). Surprisingly, we repeatedly observed that TFIIE, already detected at $t = 0$ on the RARβ2 promoter, progressively decreased in normal fibroblasts (Fig. 1h, i, Supplementary Fig. 1i, j); we however noticed a recurrent lower recruitment of TFIIE in TTD/TFIIE as well as in TTD/XPD fibroblasts.

In order to investigate whether TFIIE mutation might itself affect the expression of protein coding genes, homozygous knock-in KI-IIEβ/A150P cells were generated from the U2OS cell line by using *CRISPR/Cas9* methodology (Supplementary Fig. 2a; see "Methods"); non-mutated U2OS cells (IIEβ/WT) were used as control. We first observed that the insertion of the missense mutation c.448G>C [p.Ala150Pro] within the *GTF2E2* gene that encodes TFIIEβ was sufficient to drastically reduce the cellular levels of both α and β subunits of TFIIE complex (Supplementary Fig. 2b). After t-RA treatment, we then observed that the pattern of *RARβ2* mRNA synthesis was reduced in KI-IIEβ/A150P cells when compared with that observed in wild-type cells (Fig. 1j).

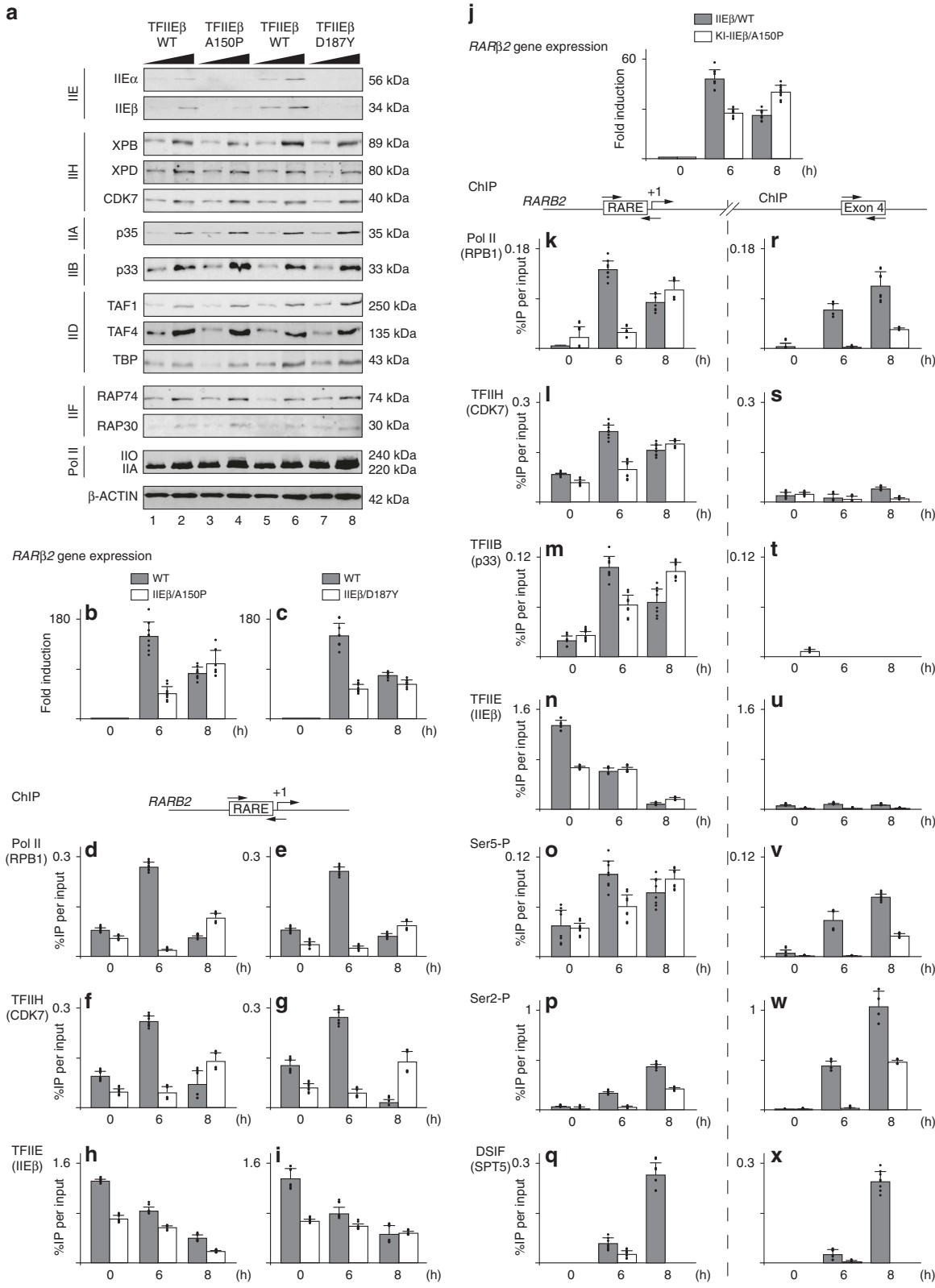

ChIP assays next revealed that the recruitment of Pol II (visualized by the presence of RPB1), TFIIH (CDK7), and TFIIB (p33) was reduced in KI-IIEβ/A150P cells when compared with that observed in IIEβ/WT cells (Fig. 1k, l, m), which correlated with the profile of RARβ2 mRNA synthesis (Fig. 1j). Interestingly, and as previously observed (Fig. 1h), TFIIE (using antibodies raised against TFIIEβ) detected at $t = 0$ was progressively removed from RARβ2 promoter in IIEβ/WT cells (Fig. 1n); its recruitment was however deficient in KI-IIEβ/A150P cells. Furthermore, an improper Ser5-P of Pol II was observed at the RARβ2 promoter (Fig. 1o). Interestingly, in IIEβ/WT and KI-IIEβ/A150P cells, the nuclear receptor RARα was similarly recruited to its response element (RARE) after 6 h of t-RA treatment (Supplementary Fig. 2c); however, RARα remained at

**Fig. 1** Defective transcription in cells harboring TFIIE mutations. **a** Increasing amounts of whole-cell lysates isolated from fibroblasts of patients with TFIIEβ/A150P, /D187Y, and of unaffected parents were used for immunoblot analysis of TFIIE (IIEα and IIEβ), TFIIH (XPB, XPD, and CDK7), TFIIA (p35), TFIIB (p33), TFIID (TAF1, TAF4, and TBP), TFIIF (RAP74 and RAP30), and Pol II (the hypo IIA- phosphorylated and hyper IIO-phosphorylated forms of RPB1). β-actin was used as loading control. **b, c** Wild-type (gray boxes) and TTD fibroblasts (open boxes) with the mutation IIEβ/A150P (panel **b**) and IIEβ/D187Y (panel **c**) have been treated with t-RA (10 μM). Relative *RARβ2* gene expression have been measured by RT-PCR after 0, 6, and 8 h of t-RA treatment (panels **b** and **c**). The mRNA levels were normalized to the 18S RNA amount. The results ($n = 9$, means ± s.d.) are presented as *n*-fold induction relative to non-treated cells. **d–i**: ChIP experiments have been done 0, 6, and 8 h post-t-RA treatment from wild-type (gray boxes) and TTD fibroblasts (open boxes) with the mutation IIEβ/A150P (panels **d**, **f**, and **h**) and IIEβ/D187Y (panels **e**, **g**, and **i**) to analyze the recruitment of Pol II (RPB1, **d** and **e**), TFIIH (CDK7, **f** and **g**) and TFIIE (TFIIEβ, **h** and **i**) at the *RARB2* proximal promoter. The values ($n = 8$, means ± s.d.) are expressed as percentage of immunoprecipitated DNA relative to the input. **j** Relative *RARβ2* gene expression after 0, 6, and 8 h of t-RA treatment in IIEβ/WT (gray boxes) and KI-IIEβ/A150P (open boxes) cells. The mRNA levels were normalized to the 18S RNA amount. The results ($n = 9$, means ± s.d.) are presented as *n*-fold induction relative to non-treated cells. **k–x** ChIP experiments have been done 0, 6, and 8 h post-t-RA treatment from IIEβ/WT (gray boxes) and KI-IIEβ/A150P (open boxes) cells. We analyzed at the RARB2 proximal promoter (panels **k–q**) and exon 4 (panels **r–x**) the recruitment of Pol II (RPB1, **k**, **r**), TFIIH (CDK7, **l**, **s**), TFIIB (p33, **m**, **t**), TFIIEβ (**n**, **u**), Ser5-P (**o**, **v**), Ser2-P (**p**, **w**), and DSIF (SPT5, **q**, **x**). The values ($n = 8$, means ± s.d.) are expressed as percentage of immunoprecipitated (IP) DNA relative to the input. Source data are provided as a Source Data file

the promoter 8 h post-treatment in KI-IIEβ/A150P cells, a situation not observed in normal cells. Moreover, the phosphorylation by CDK7 of the nuclear receptor RARα (which is a prerequisite for accurate expression of its corresponding target genes)[22] was not modified in t-RA-treated KI-IEβ/A150P cells (Supplementary Fig. 2d). These results suggest that the disruption of the RARβ2 mRNA synthesis did not result from deficiency of the RARα-activation process, but rather from defects during PIC assembly.

We also investigated the possible consequences of a defective PIC formation for ongoing transcription. ChIP assays at exon 4 showed increased phosphorylation of elongating Pol II at Ser2 (Ser2-P) in IIEβ/WT cells after 6 h t-RA treatment (Fig. 1w). Persistence of phosphorylated elongating Pol II at 8 h, when RARβ2 mRNA tends to decrease, might be due to the cyclic profile of the *RARβ2* gene expression[30] and the very distal position of exon 4 from the transcription start site (~140 kb). Strikingly, Pol II phosphorylation was strongly disrupted in KI-IIEβ/A150P cells (Fig. 1w). It is worthwhile to notice that under our experimental conditions, neither TFIIB nor TFIIE (TFIIEβ) were detected at exon 4 in normal and mutated cells (Fig. 1t, u); only background levels were detected for CDK7 (Fig. 1s). Finally, the recruitment of the transcription elongation factor DSIF[31] (visualized via its SPT5 subunit) was clearly impaired in mutated cells (Fig. 1x), suggesting that the previously observed alterations in PIC formation influence transcriptional elongation. All together our data show how the arrival of the transcriptional machinery with Pol II and the release of TFIIE occur at the promoter of activated gene to engage RNA synthesis, and how TFIIE and TFIIH mutations notably disturb such coordinated events.

**Mutations within TFIIEβ disrupt transcription initiation**. We next asked how the TFIIE/TFIIH partnership occurred. Recombinant TFIIE (rIIE) containing rIIEα/WT subunit with either rIIEβ/WT, /A150P, or /D187Y were produced and purified (Fig. 2a). We first observed that addition of increasing amounts of either rIIEαβ/WT, rIIEαβ/A150P, or rIIEαβ/D187Y in NER reaction (containing XPC, TFIIH, XPA, RPA, XPF/ERCC1, and XPG as well as a cisplatinated substrate)[32], did not influence the removal of the damaged oligonucleotides (Supplementary Fig. 3a). On the contrary, purified recombinants rIIH-XPD/R112H or rIIH-XPD/R722W (Fig. 2b) drastically prevented NER (Supplementary Fig. 3b). This strongly suggested that TFIIE did not modulate in vitro NER activity.

We next investigated the consequences of TFIIE mutations during in vitro transcription assays. The rIIEs were incubated together with the adenovirus major late promoter (AdMLP, run-

off of 309nt), TFIIA, TFIIB, TFIID (TBP), TFIIF, and TFIIH as well as Pol II[12]. We observed that 5, 10, and 20 min after addition of nucleoside triphosphates (NTPs, including radiolabelled CTP), rIIEαβ/A150P and /D187Y led to a lower RNA synthesis than rIIEαβ/WT (Fig. 2c). The defect was slightly more pronounced with rIIEαβ/A150P than /D187Y (compare lanes 5–7 to 8–10). Similarly, transcriptional defect was observed in the presence of rIIH-XPD/R112H or /R722W (Fig. 2d, lanes 5–7 and 8–10, respectively).

To evaluate whether TFIIE mutations might interfere with the first steps of transcription, we performed in vitro abortive initiation assays (Fig. 2e), during which we measured trinucleotide synthesis initiated by a reporter priming dinucleotide RNA (CpA) hybridized to the transcription start site of the AdMLP[33,34]. Upon addition of radiolabelled CTP and increasing amounts of either rIIEαβ/A150P or /D187Y, trinucleotide (CpApC) synthesis was impaired (lanes 5–7 and 8–10). The defect was greater with rIIEαβ/A150P than /D187Y, which paralleled what was observed during in vitro run-off assays (Fig. 2c). Interestingly, the difference between the two mutated forms of TFIIE was more pronounced in run-off than in abortive assays, suggesting that early transcriptional steps might be differently affected by rIIEαβ/A150P and rIIEαβ/D187Y; the latter ensures initiation but may fail at subsequent steps. Finally, impairment of trinucleotide synthesis was also observed when increasing amounts of rIIH-XPD/R112H or /R722W together with rIIEαβ/WT were used (Fig. 2f).

Together, the above results underlined the essential role of TFIIE during transcription initiation. In addition, it seems that TTD-related mutations within TFIIE and TFIIH both disrupt the early steps of transcription initiation.

**TFIIEα is essential to incorporate the CAK within the PIC**. We then investigated the role of TFIIE during PIC formation. Biotinylated AdMLP was immobilized to streptavidin beads and incubated with all the GTFs (TFIIA, TFIIB, TBP, TFIIF, TFIIH), Pol II, and either rIIEαβ/WT, rIIEαβ/A150P, or rIIEαβ/D187Y (Fig. 3a). After several washes, western blot analysis of the remaining proteins bound to DNA showed that Pol II (revealed by its RPB2 subunit), TFIIF (RAP74), and TBP were associated with DNA in the absence of TFIIE (lane 2); the signals corresponding to the Core-TFIIH (XPB) and the XPD bridging component of TFIIH were however reduced, with an even more severe for CAK (CDK7). Interestingly, addition of rIIEαβ/WT promoted the recruitment of TFIIH (lane 3), consistent with the observation that TFIIH co-immunoprecipitated with rIIEαβ/WT (Fig. 3b, lanes 1 and 2). Strikingly, rIIEαβ/A150P and /D187Y reduced the binding of CAK (CDK7) (Fig. 3a, compare lanes 4

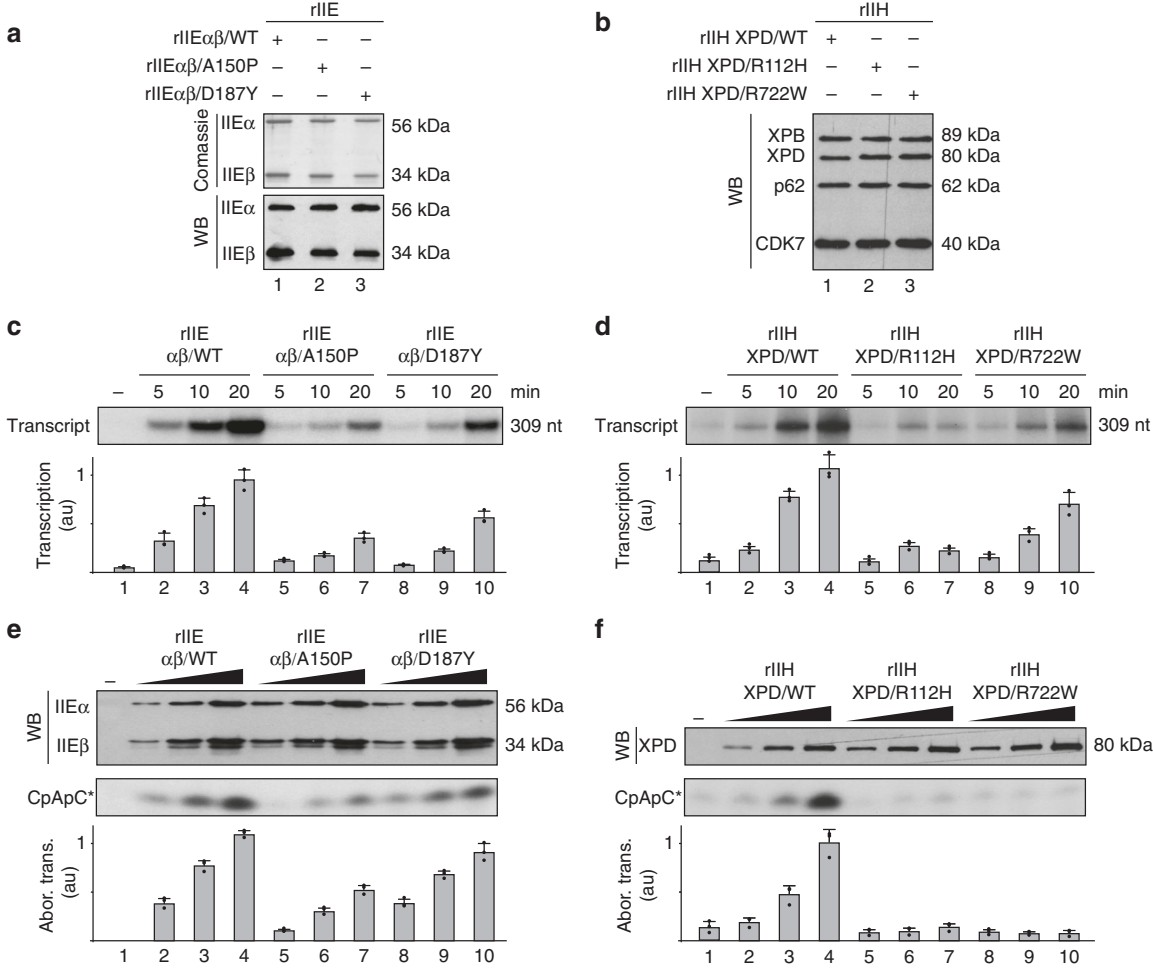

**Fig. 2** Defective TFIIE and TFIIH affect transcription initiation. **a** Production of the recombinant mutated forms of TFIIE (rIIE). Equal amounts of purified rIIE were resolved by SDS–PAGE. Coomassie blue staining gel (top panel) and western blot (WB, bottom panel) with antibodies raised against TFIIEα and TFIIEβ are shown. **b** Production of the recombinant mutated forms of TFIIH (rIIH). Equal amounts of purified rIIHs were resolved by SDS–PAGE and immunoblotted for XPB, XPD, p62, and CDK7. **c**, **d** Purified rIIEs (**c**) and rIIHs (**d**) were added to in vitro reconstituted transcription systems, as indicated. After addition of NTPs (including radiolabelled CTP), the transcription activity was assessed after 5, 10, and 20 min of incubation. The length of the corresponding transcript (309nt) is indicated on the right side. The signals were quantified ($n = 3$, means ± s.d.) and plotted in arbitrary units (a.u.). The results are representative of three independent experiments. **e**, **f** Abortive transcription assays using rIIEs (**e**) and rIIHs (**f**). After preinitiation complex assembly using equivalent amounts of the different purified rIIEs and rIIHs (as revealed by western blots), phosphodiester bond synthesis was initiated by the addition of priming dinucleotides CpA and radiolabelled CTP. After 30 min of incubation, the radioactive trinucleotide (CpApC*) synthesis was stopped, quantified ($n = 3$, means ± s.d.) and plotted in arbitrary units (a.u.). The results are representative of two independent experiments. Source data are provided as a Source Data file

and 5 with lane 3 and histogram), related to the lower capacity of the mutated rIIEs to entirely co-immunoprecipitate with TFIIH (Fig. 3b, lanes 3–6).

The above observations prompted us to explore the involvement of TFIIE subunits in the recruitment of TFIIH within the PIC. In the absence of rIIEβ/WT, rIIEα/WT was unable to bind the AdMLP/Pol II/TBP/TFIIA/TFIIB/TFIIF complex (Fig. 3c, lane 2). Strikingly, whereas TFIIH interacted with rIIEβ/WT rather than rIIEα/WT in solution (Fig. 3d, lanes 2 and 4), both TFIIE subunits were required to fully recruit entire TFIIH, including CAK, to the AdMLP (Fig. 3c, lane 7). These observations may be related to the ability of rIIEαβ/WT to interact with TFIIH (Fig. 3b), in particular with the CAK (Supplementary Fig. 4)[35]. Coimmunoprecipitation between TFIIE and TFIIH components were slightly reduced in the presence of rIIEαβ/A150P or rIIαβ/D187Y variants (Fig. 3b, lanes 3–6). Interestingly, rIIEβ mutations weakened interaction with rIIEα/WT, as observed at high salt concentration (Fig. 3e), possibly due

to the position of the TFIIEβ mutations in the WH2 domain which interacts with TFIIEα[36,37]. For their part, XPD/TTD mutations affected the integrity of TFIIH by destabilizing the interaction between the CAK (CDK7) and the Core (XPB) sub-complexes (Fig. 3f). As a consequence, the recruitment of the CAK at the AdMLP was reduced, as observed for XPD/R722W (Fig. 3g, compare lanes 5 to 3).

Altogether, the above data underlined the role played by TFIIEβ to anchor TFIIEα, which further allows incorporation of the whole TFIIH within the PIC. Additionally, we showed that mutations in TFIIE and TFIIH disturb the integration of these factors within the PIC.

**ATP promotes the release of TFIIEα and the CAK.** We next investigated how TFIIE and TFIIH cooperate to phosphorylate Pol II and to open DNA during transcription[13,38]. Addition of ATP to the PIC formation assay (containing the immobilized

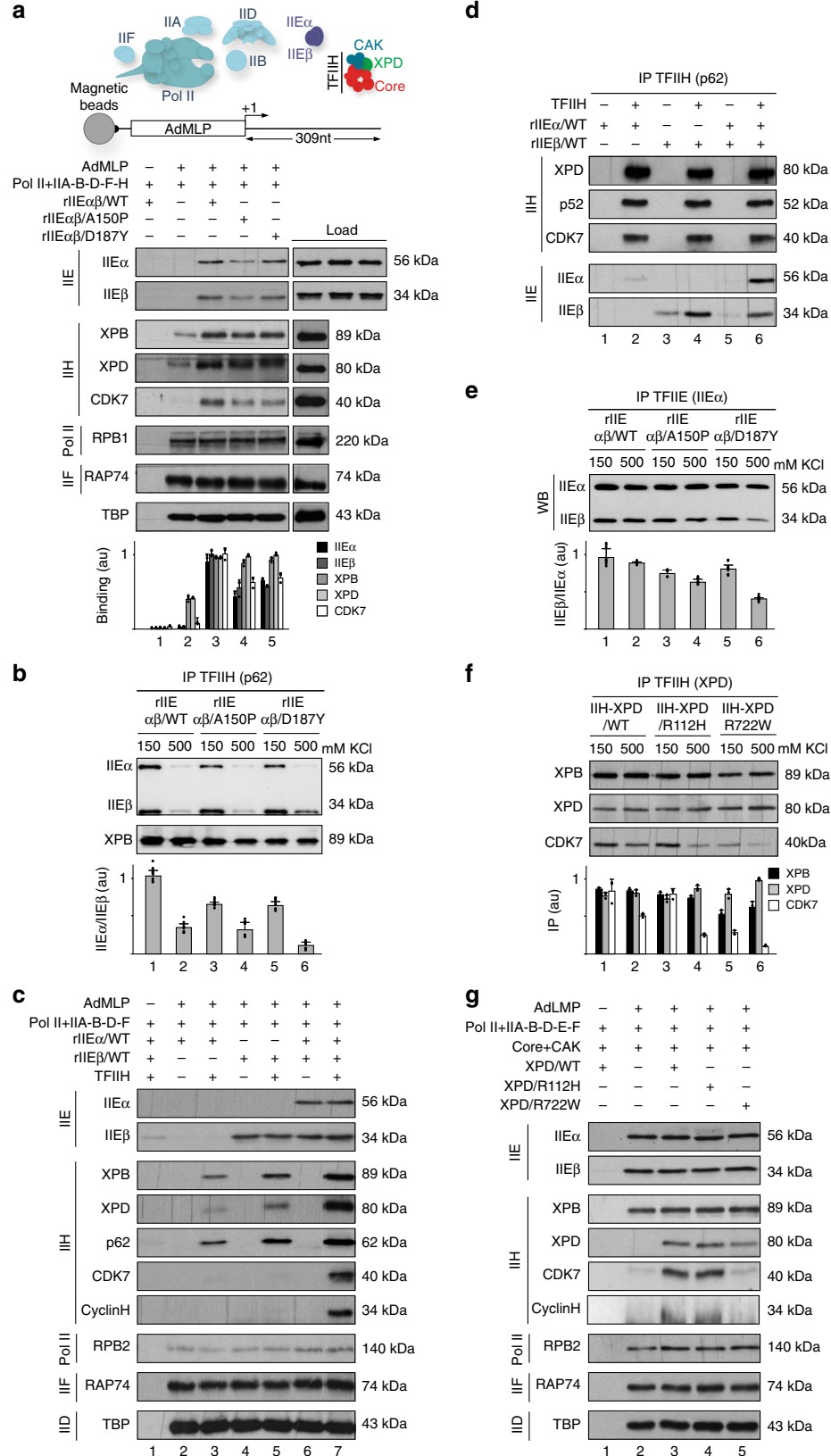

AdMLP, Pol II, and all the GTFs including rIIEαβ/WT) led to accumulation of Ser5-P of Pol II (Fig. 4a, lane 3). Furthermore, in such conditions, the CAK (visualized by CDK7 and Cyclin H) and TFIIEα were separated from the Core-TFIIH and TFIIEβ, which were maintained at the DNA template together with Pol II (RPB2) and TFIIF (RAP74). In presence of ATP-γS, whose hydrolysis is very low compared to ATP, neither Pol II phosphorylation nor the release of the CAK and TFIIEα occurred (lane 4). Interestingly, Pol II phosphorylation and TFIIEα release were not observed in absence of the CAK (Fig. 4b, lane 3). Immobilized AdMLP template was next incubated with Pol II and all the GTFs including rIIEαβ/WT and rIIH/WT in presence of

**Fig. 3** Defective TFIIE and TFIIH alter the preinitiation complex. **a** Biotinylated AdMLP bound to streptavidin magnetic beads was incubated with Pol II, TFIIA, TFIIB, TFIID (TBP), TFIIF, and TFIIH, in the presence (+) of either rIIEα/WT, rIIEα/A150P, or rIIEα/D187Y. After washes, the binding of different factors was evaluated by immunoblotting. The loading proteins used for PIC formation and the different rIIEs (in the following order rIIEαβ/WT, rIIEαβ/A150P, and rIIEαβ/D187Y) are indicated at the right part of the panel (load). The signals for IIEα, IIEβ, XPB, XPD, and CDK7 were quantified ($n = 3$, means ± s.d.) and plotted in arbitrary units (a.u.). The results are representative of three independent experiments. **b** Immunoprecipitated rIIH/WT (using anti-p62) was incubated with rIIEs at 150 and 500 mM KCl. After washes, the coimmunoprecipitated proteins were resolved by SDS–PAGE and blotted with anti-XPB, anti-IIEα, and anti-IIEβ. Graph shows in arbitrary units (a.u.) the ratio IIEα/IIEβ for each rIIE. The results are representative of three independent experiments. **c** Pol II, TFIIA, TFIIB, TFIID (TBP), and TFIIF were incubated with biotinylated AdMLP bound to streptavidin beads in presence (+) of rIIEα/WT, rIIEβ/WT, and TFIIH. Immunoblot analysis has been done with different antibodies, as indicated. **d** TFIIH was immunoprecipitated with anti-p62 and incubated (+) with rIIEα/WT and/or rIIEβ/WT. After washes, immunoblot analysis of TFIIH (XPD, p52, CDK7) and TFIIE (IIEα and IIEβ) has been done. **e** Recombinant rIIEα/WT, co-produced with either rIIEβ/WT (lanes 1 and 2), rIIEβ/A150P (lanes 3 and 4) or rIIEβ/D187Y (lanes 5 and 6), was immunoprecipitated at 150 and 500 mM KCl. After washes, the proteins were resolved by SDS–PAGE and blotted with antibodies against IIEα and IIEβ. Graph shows the ratio IIEβ/IIEα ($n = 3$, means ± s.d.) in arbitrary units (a.u.). The results are representative of three independent experiments. **f** rIIH containing either XPD/WT (lanes 1 and 2), XPD/R112H (lanes 3 and 4), or XPD/R722W (lanes 5 and 6) were immunoprecipitated with anti-XPD and incubated at 150 and 500 mM KCl. After washes, the bound proteins were resolved by SDS–PAGE. The immunoprecipitated signals (IP) for XPB, XPD, and CDK7 were quantified ($n = 3$, means ± s.d.) and plotted in arbitrary units (a.u.). The results are representative of three independent experiments. **g** Streptavidin magnetic beads were incubated (when indicated, +) with biotinylated AdMLP, Pol II, TFIIA, TFIIB, TFIID (TBP), rIIEαβ/WT, TFIIF, the Core-TFIIH, the CAK and either XPD/WT, /R112H, or /R722W. After washes, immunoblot analysis of different factors has been done. The results ($n = 3$, means ± s.d.) are representative of three independent experiments. Source data are provided as a Source Data file

ATP and THZ1, a specific CDK7 inhibitor that covalently targets a cysteine residue located outside of the canonical kinase domain[39]. In these conditions, Pol II remained unphosphorylated and both TFIIEα and the CAK were maintained at the DNA template (Fig. 4c, lanes 4 and 5). Taken together, the above data strongly suggested that once Pol II is phosphorylated by CDK7, TFIIEα as well as the CAK are no longer required at the promoter.

We next examined whether promoter opening was required for Pol II phosphorylation and the release of TFIIEα and the CAK. In our in vitro assays, the addition of triptolide (TPL), a chemical compound known to inhibit the ATPase activity of XPB and to block transcription (Supplementary Fig. 5a)[40], did not modify the ATP-dependent release of the CAK and TFIIEα (Fig. 4d). Furthermore, Ser5-P of Pol II was maintained (lane 5). Similarly, we also observed that the release of the CAK and TFIIEα still occurred in the presence of the mutated form rIIH-XPB/Fs740 (Fig. 4e, lanes 4 and 5), which prevented the XPB activity and affected transcription (Supplementary Fig. 5b)[41]. As a further control, we also found that rIIH-XPB/K346R, which harbors a mutation in the ATPase Walker motif that prevents ATP hydrolysis and disrupts transcription (Supplementary Fig. 5c)[42,43], did not prevent the Pol II phosphorylation, as well as the release of the CAK and TFIIEα (Fig. 4f, lanes 4 and 5). Altogether, the above data suggested that promoter opening was not a prerequisite for Pol II phosphorylation nor the release of the CAK and TFIIEα.

**The CAK is released during RNA synthesis**. We next tested whether the PIC was able to promote RNA synthesis after having released the CAK and TFIIEα. After incubation of the immobilized AdMLP template with Pol II and all the GTFs, we added ATP as indicated (Fig. 5a). After washes, the remaining complex (containing the IIO-phosphorylated form of RPB1, lane 3) was incubated with all the NTPs to allow RNA synthesis (lane 4, 309nt run-off transcript). In such conditions, we observed a release of the CAK (as vizualized by CyclinH) and TFIIEα; the Core-TFIIH was partially released (as vizualized by XPB) while TFIIEβ was maintained at the promoter. It is noteworthy that less RNA synthesis occurred after an ATP pre-incubation was done (Fig. 5b), which was in agreement with previous observations showing that the activity of PICs was reduced by exposure to ATP alone[44,45]. Since TFIIE can reactivate Pol II after ATP pre-incubation[46], we supplemented with increasing amounts of IIEα

following ATP treatment, washes, and addition of NTPs (Fig. 5c). This increased the presence of TFIIEα at the promoter (lanes 5 and 6), and RNA synthesis (Fig. 5d, lanes 4 and 5), as previously observed[46].

We next explored the fate of TFIIEβ during transcription. To determine whether additional factors might contribute to its release, the immobilized in vitro assay was supplemented with whole cell extracts (WCE, isolated from HeLa cells) in the presence of NTPs (Fig. 5e, f). While RNA synthesis occurred, we observed the complete removal of the Core-TFIIH (lane 3). Remarkably, the TFIIEβ subunit was also released, concomitant with the recruitment of the elongation factor DSIF (SPT5), as observed during ChIP experiments (Fig. 1q). TFIIEβ was partially removed in the presence of ATP alone (Fig. 5f, lane 3). However, the elongation factor DSIF (SPT5) was not recruited when compared to what obtained in presence of NTP (lane 4), suggesting that ATP addition was sufficient for TFIIEα eviction but that all four NTPs were needed for complete removal of TFIIEβ and DSIF recruitment.

Taken together, these results strongly suggested that TFIIEα and the CAK, followed by the Core-TFIIH and TFIIEβ, are sequentially released from the promoter while elongation factors take place to pursue transcription.

**TTD mutations prevent CAK recruitment**. We next investigated how TTD mutations might perturb the transcription initiation process. Immobilized AdMLP was first incubated with Pol II, the GTFs (TFIIA, TFIIB, TBP, TFIIF, TFIIH) and either rIIEαβ/WT, rIIEαβ/A150P, or rIIEαβ/D187Y in the presence of ATP (Fig. 6a). After washes, western blots showed that the ATP-dependent release of the CAK also occurred in presence of rIIEαβ/A150P and rIIEαβ/D187Y (lanes 4 and 6); it should be however noted that the CAK was recruited much less in the absence of ATP (lanes 3 and 5) as was previously observed (Fig. 3a). Similarly, the CAK was released upon ATP addition in the presence of TFIIH XPD/WT (Fig. 6b, lanes 2 and 3), XPD/R112H (lanes 4 and 5) and to some extent with XPD/R722W (lane 6 and 7). Consequently, TTD-causative mutations impaired the recruitment of the CAK subcomplex on transcription start sites.

In vitro transcription assays were also performed in the presence of the basal transcription machinery with the different rIIEs and rIIHs variants (Fig. 6c, d). Reactions were stopped 5, 10, and 20 min after addition of cold NTPs and analyzed by western blot using antibodies raised against Ser5-P of Pol II. The presence

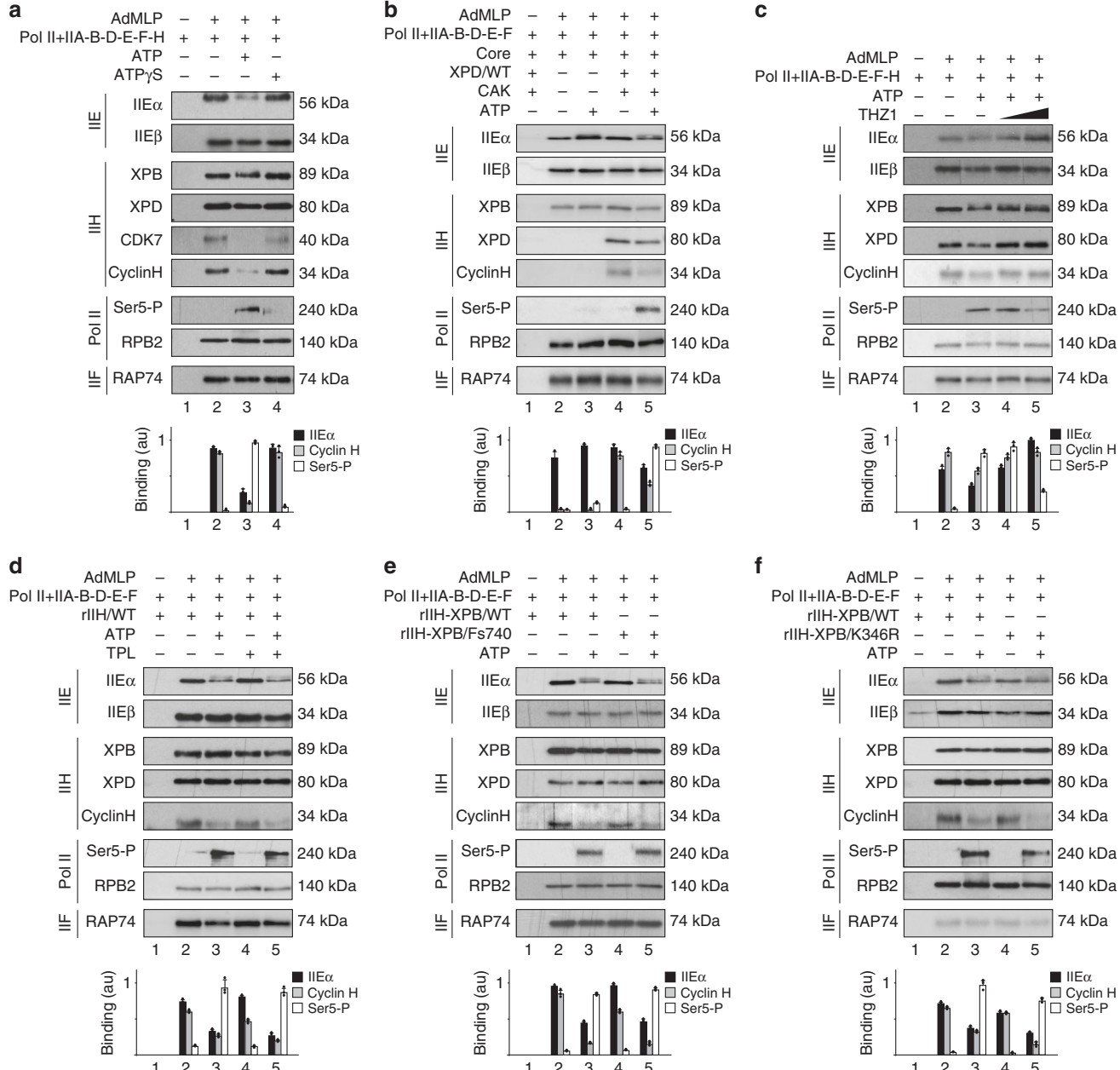

**Fig. 4** TFIIEα and CAK releases and Pol II phosphorylation occur before DNA opening. **a** Streptavidin magnetic beads were incubated (+) with biotinylated AdMLP, Pol II, TFIIA, TFIIB, TFIID (TBP), rIIEαβ/WT, TFIIF and rIIH/WT, in presence of either ATP (200 μM) or ATPγS (200 μM). After washes, the binding of various factors has been evaluated by immunoblotting. The immunoblot signals for TFIIEα, CyclinH, and Ser5-P were quantified (n = 3, means ± s.d.) and plotted in arbitrary units (a.u.). The results are representative of three independent experiments. **b** Streptavidin magnetic beads were incubated (when indicated, +) with biotinylated AdMLP, Pol II, TFIIA, TFIIB, TFIID (TBP), rIIEαβ/WT, TFIIF, the Core-TFIIH, XPD/WT, and the CAK, in presence of ATP (200 μM). After washes, immunoblot analysis was done for different proteins. The immunoblot signals for TFIIEα, CyclinH, and Ser5-P were quantified (n = 3, means ± s.d.) and plotted in arbitrary units (a.u.). The results are representative of three independent experiments. **c** Streptavidin magnetic beads were incubated (+) with biotinylated AdMLP, Pol II, TFIIA, TFIIB, TFIID (TBP), rIIEαβ/WT, TFIIF, and rIIH/WT, in presence of ATP (200 μM) and THZ1 (1 and 10 μM). The immunoblot signals for TFIIEα, CyclinH, and Ser5-P were quantified (n = 3, means ± s.d.) and plotted in arbitrary units (a.u.). The results are representative of three independent experiments. **d** When indicated (+), biotinylated AdMLP bound to streptavidin magnetic beads were preincubated 20min (at 25 °C) with the general transcription machinery (Pol II, TFIIA, TFIIB, TFIID (TBP), rIIEαβ/WT, TFIIF, and rIIH/WT) in presence of Triptolide (TPL, 10 μM) before addition of ATP (200 μM). The immunoblot signals for TFIIEα, CyclinH, and Ser5-P were quantified (n = 3, means ± s.d.) and plotted in arbitrary units (a.u.). The results are representative of three independent experiments. **e** Streptavidin magnetic beads were incubated (+) with biotinylated AdMLP, Pol II, TFIIA, TFIIB, TFIID (TBP), rIIEαβ/WT, and TFIIF, in presence of ATP (200 μM) and either rIIH-XPB/WT or /Fs740. The immunoblot signals for TFIIEα, CyclinH, and Ser5-P were quantified (n = 3, means ± s.d.) and plotted in arbitrary units (a.u.). The results are representative of three independent experiments. **f** Streptavidin magnetic beads were incubated (+) with biotinylated AdMLP, Pol II, TFIIA, TFIIB, TFIID (TBP), rIIEαβ/WT, and TFIIF, in presence of ATP (200 μM) and either rIIH-XPB/WT or /K346R. The immunoblot signals for TFIIEα, CyclinH, and Ser5-P were quantified (n = 3, means ± s.d.) and plotted in arbitrary units (a.u.). The results are representative of three independent experiments. Source data are provided as a Source Data file

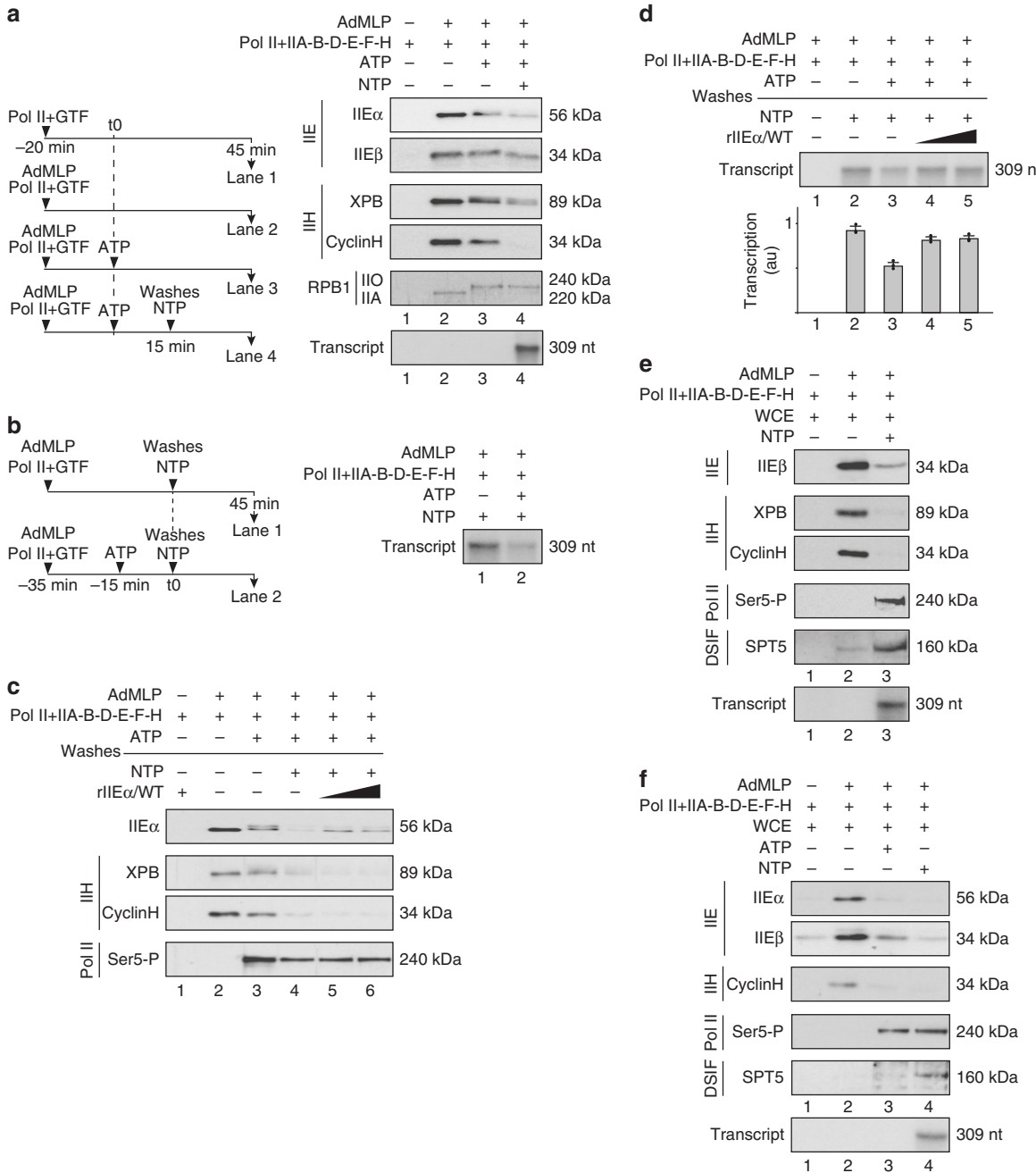

**Fig. 5** The CAK is released during RNA synthesis. **a** Streptavidin magnetic beads were incubated (as depicted in the scheme, left part) with biotinylated AdMLP, Pol II, and the GTFs (TFIIA, TFIIB, TFIID (TBP), rIIEαβ/WT, TFIIF, and TFIIH/WT), in presence of ATP (200 μM). After 15 min of incubation with ATP, the beads were washed and NTPs (200 μM) were added (lane 4). Immunoblot analysis (for TFIIEα, TFIIEβ, XPB, CyclinH, RPB1) and the transcription activity (309nt run-off transcript) were assessed after 45 min of incubation. The results are representative of three independent experiments. **b** Streptavidin magnetic beads with biotinylated AdMLP, Pol II, and the GTFs (TFIIA, TFIIB, TFIID (TBP), rIIEαβ/WT, TFIIF, and TFIIH/WT) were pre incubated 15 min (as depicted in the scheme) in presence (lane 2) or absence (lane 1) of ATP (200 μM). The beads were washed and NTPs (200 μM) were added. Transcription activity (309nt run-off transcript) were assessed after 45 min of incubation. The results are representative of three independent experiments. **c**, **d** Streptavidin magnetic beads were pre incubated (when indicated, +) with biotinylated AdMLP, Pol II and the GTFs (TFIIA, TFIIB, TFIID (TBP), rIIEαβ/WT, TFIIF, and TFIIH/WT) in presence of ATP (200 μM). After washes, NTP (200 μM) and increasing amounts of rIIEα/WT were added. Immunoblot analysis (**c**) and transcription activity (**d**) were assessed after 45 min of incubation. The transcription activities were quantified (n = 3, means ± s.d.) and plotted in arbitrary units (a.u.). The results are representative of two independent experiments. **e**, **f** Streptavidin magnetic beads were incubated (when indicated, +) with biotinylated AdMLP, Pol II, TFIIA, TFIIB, TFIID (TBP), rIIEαβ/WT, TFIIF, and TFIIH/WT, in presence of whole cell extracts (WCE), ATP (200 μM) (**f**) and NTP (200 μM) (**e**, **f**). Immunoblot analysis (for TFIIEα, TFIIEβ, XPB, CyclinH, Ser5-P, and SPT5) and the transcription activity (309nt run-off transcript) were assessed after 45 min of incubation. The results are representative of three independent experiments. Source data are provided as a Source Data file

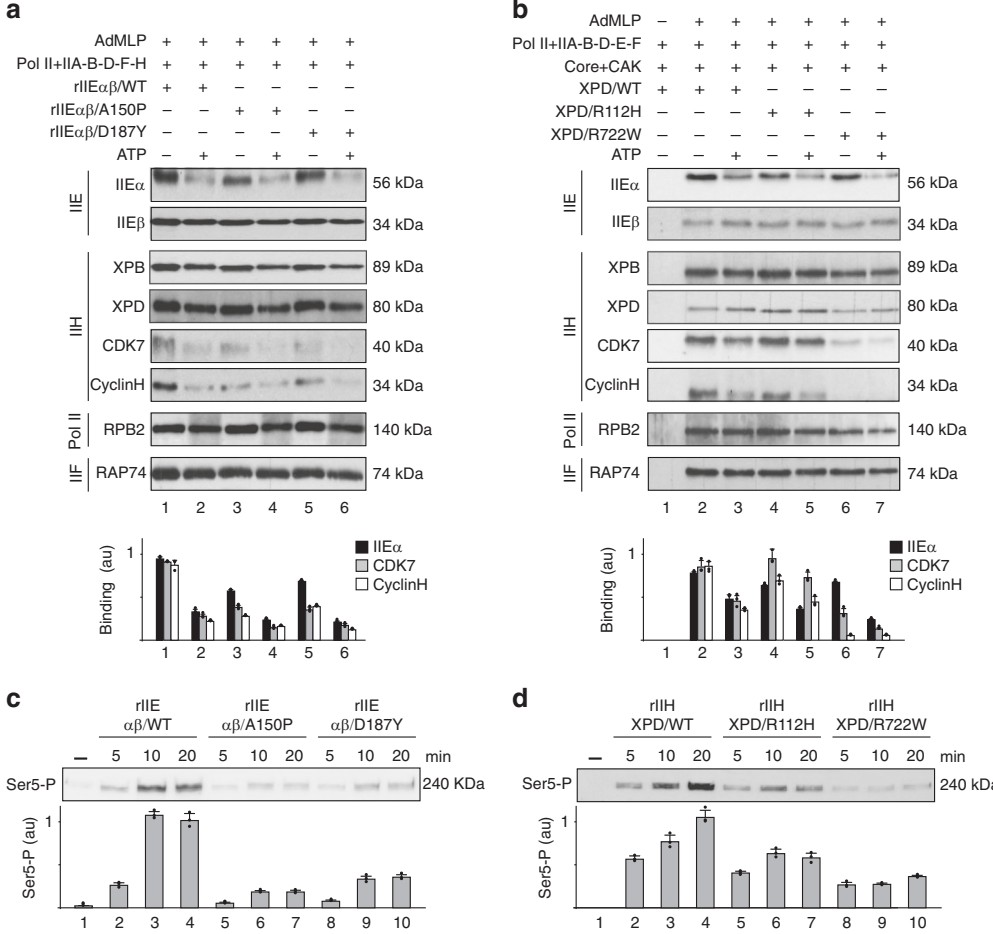

**Fig. 6** TTD mutations alter CAK release and Pol II phosphorylation. **a** Biotinylated AdMLP bound to streptavidin magnetic beads was incubated with Pol II, TFIIA, TFIIB, TFIID (TBP), TFIIF, and TFIIH/WT, in presence (when indicated, +) of ATP (200 μM) and either rIIEαβ/WT, /A150P, or /D187Y. The immunoblot signals for TFIIEα, CDK7, and CyclinH were quantified and plotted in arbitrary units (a.u.). The results (n = 3, means ± s.d.) are representative of three independent experiments. **b** Streptavidin magnetic beads were incubated (when indicated, +) with biotinylated AdMLP, Pol II, TFIIA, TFIIB, TFIID, rIIEαβ/WT, TFIIF, the Core-TFIIH, and the CAK, in presence of ATP (200 μM) and either XPD/WT, /R112H or /R722W. The immunoblot signals for TFIIEα, CDK7, and CyclinH were quantified (n = 3, means ± s.d.) and plotted in arbitrary units (a.u.). The results are representative of three independent experiments. **c**, **d** In vitro reconstituted transcription systems with rIIEs (**c**) and rIIHs (**d**) have been performed in presence of cold NTPs. Immunoblot analysis next has been done using specific antibody against Ser5-P of Pol II. Signals were quantified (n = 3, means ± s.d.) and plotted in arbitrary units (a.u.). The results are representative of three independent experiments. Source data are provided as a Source Data file

of rIIEβ/A150P and /D187Y strongly prevented CAK-dependent Pol II phosphorylation (Fig. 6c, lanes 5–7 and 8–10, respectively). Diminished Pol II phosphorylation was also observed with mutated forms of rIIH (Fig. 6d), the defect being more pronounced for XPD/R722W than XPD/R112H (lanes 5–7 and 8–10, respectively).

Taken together, these results suggested that TFIIEβ and XPD mutations cause in common a perturbation of Pol II phosphorylation by the CAK.

## Discussion

The present study aimed to dissect the interplay between TFIIE and TFIIH during transcription, from their integration within the PIC until their release before the upcoming recruitment of elongation factors. This work also revealed how TTD mutations within TFIIE and TFIIH disrupt such processes, resulting in a decrease in transcription.

Upon gene activation, the GTFs target the promoter in a sequential order, which is initiated by the recruitment of TFIID, TFIIA, TFIIB, and Pol II together with TFIIF[1,2], the whole forming a structure ready to incorporate TFIIE and TFIIH

(Fig. 7). Using immobilized template and highly purified recombinant GTFs, we show that the TFIIEα subunit is recruited at the promoter via TFIIEβ, the latter interacting with Pol II (Supplementary Fig. 6a)[37]. While the Core-TFIIH can target the PIC in absence of TFIIE, through partnerships with Pol II (Supplementary Fig. 6b), we demonstrate that TFIIEα promotes the incorporation of the CAK module of TFIIH within the PIC (Fig. 3c), which is in accordance with structural analyses[35]. The PIC formation assays questions about the retention, and loss, of transcriptional components while Pol II advances to transcribe and is retained on the DNA template (Fig. 5a, lane 4). In a reconstituted in vitro transcription assay, we cannot exclude Pol II retention at the end of the DNA template, a situation requiring additional factors for the release of the polymerase. Furthermore, formation of inactive complexes can occur, resulting from incorrectly folded proteins and/or incomplete PIC assembly. However, in our experimental conditions, almost all the Pol II was phosphorylated after ATP addition, which is illustrated by the complete conversion of RNA polymerase IIA to IIO (Fig. 5a, lane 3). This suggests that the majority of the polymerase bound to the DNA template may be integrated into complexes that allow

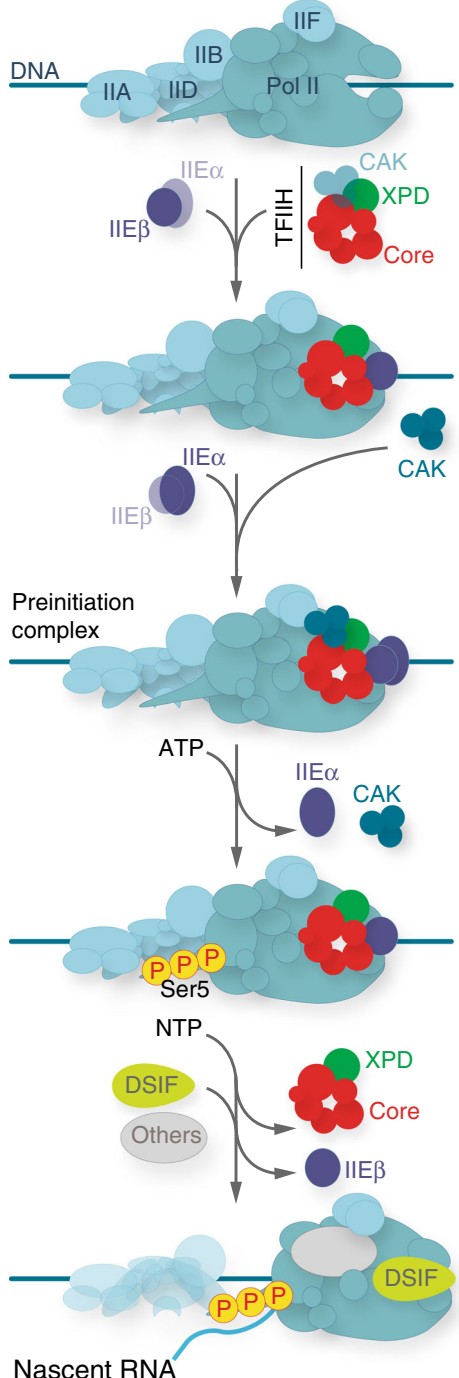

**Fig. 7** Sequential association/dissociation of TFIIE and TFIIH. Once a stable TFIID–TFIIA–TFIIB–Pol II–TFIIF promoter complex is assembled, TFIIE and TFIIH can be recruited. The TFIIEβ subunit is required to anchor TFIIEα. Altgough the Core-TFIIH sub-complex can incorporate the complex in absence of TFIIE, TFIIEα, and TFIIEβ are both required to anchor the CAK within the PIC. In presence of ATP, Pol II phosphorylation is accompanied by the release of the CAK and TFIIEα from the promoter, a process that takes place before DNA opening. In presence of NTP, RNA synthesis is initiated, the Core-TFIIH and TFIIEβ are removed while DSIF and other elongation factors are recruited

it to become active. Interestingly, a selective (i.e. TFIIE and TFIIH) but not complete removal of GTFs was observed once Pol II is engaged in elongation (lane 4), as previously suggested[47]. Such dynamic recruitment/release of factors illustrates the

complexity of the sequential events that are required to accomplish promoter clearance[48].

Many questions remain unanswered about the fate of TFIIE and TFIIH during transcription initiation, their respective contributions being debated for a number of years[6,33]. Here we show that TFIIE allows the integration of the CAK within the PIC (Fig. 3c), leading to Pol II phosphorylation by CDK7 (Fig. 4a)[9]. Once Pol II is phosphorylated, TFIIEα and the CAK are released from the promoter (Fig. 4a–c). Interestingly, the CAK dissociation observed during transcription mirrors what happens during NER. Indeed, upon arrival of the NER factor XPA on the damaged DNA (already targeted by XPC and TFIIH), the CAK is released in an ATP-dependent manner from the Core-TFIIH[49].

Our results show that Pol II phosphorylation, and release of TFIIEα and the CAK, do not require DNA opening (Fig. 4d–f), since these events are not impeded by TPL that inhibits XPB enzymatic activity[40], or mutations in XPB that affect its promoter opening activity[41–43].

The ATP-dependent release of TFIIEα and the CAK raises the question of the fate of the Core-TFIIH and TFIIEβ within the remaining complex. The Core-TFIIH is required for next transcriptional events; in particular, the translocase activity of XPB allows DNA pumping into the active site cleft of Pol II[15,50]. Interestingly, the Core-TFIIH is not longer associated with the promoter once RNA synthesis is undertaken (Fig. 5a). Concerning TFIIEβ, this subunit interacts with DNA (Supplementary Fig. 7), which might prove useful during promoter opening[6,7,51]. TFIIEβ is also no longer maintained at the promoter (Figs. 1h, i, n, 5a and Supplementary Fig. 1i, j); its release parallels the recruitment of elongation factors, such as DSIF (Figs. 1q, 5e, f), a heterodimer of SPT4 and SPT5 that mediates pausing by Pol II early after initiation[31]. Remarkably, structural analyses showed that the stalk domain of Pol II binds TFIIE as well as DSIF[52]. The fact that DSIF and TFIIE bind overlapping sites in the clamp region of Pol II suggest that DSIF might substitute for TFIIE[53–55], a situation strongly corroborated by our results (Fig. 5e). Interestingly, reductions in DSIF and increases in TFIIE near promoters have been described as a result of CDK7 inhibition[56–58]. Whether or not the eviction of TFIIE requires post translational modifications remains to be determined. In this regard, it should be pointed out that a doublet for TFIIEα was observed in presence of ATP (Figs. 4, 5 and 6). This might result from the phosphorylation of TFIIEα, likely by CDK7, as previously observed[9,56]. Although such apparent TFIIEα phosphorylation is of interest, substantial investigations should be undertaken to identify the phosphorylation(s) sites(s) and to determine the function(s) of such modification(s) during transcription.

The TFIIEβ/A150P and /D187Y mutations have been recently associated with TTD syndrome[25,26], which is so far mainly related to TFIIH mutations. The results presented here show that TTD-related mutations within TFIIE and TFIIH have in common (i) decreasing the cellular concentration of these complexes (Fig. 1a and Supplementary Fig. 1b) by weakening their stability (Fig. 3e, f), (ii) affecting the anchoring of the CAK within the PIC (Figs. 3a, g and 6a, b), (iii) hampering Pol II phosphorylation (Fig. 6c, d) and therefore (iv) disrupting transcription in an in vitro (Fig. 2c, d) and ex vivo context (Fig. 1b, c, j, Supplementary Fig. 1c, d). Consequently, mutations in either *GTF2E2* (which affect the interaction of TFIIEβ with TFIIEα, Fig. 3e) or *XPD* (which disturb the TFIIH stability, Fig. 3f) share similar consequences, mainly the lack of recruitment of the CAK and of its activity during transcription. The fact that mutations in TFIIE do not alter NER (Supplementary Fig. 3a) support the idea that Trichothiodystrophy is mainly related to transcription deficiencies. Overall, TTD-related mutations in TFIIE and TFIIH lead to numerous similar impacts during transcription initiation,

which might explain in part why mutations within two complexes can lead to the same transcriptional syndrome.

## Methods

**Reagents and resources**. The key reagents and resources (antibodies, chemical, cell lines, oligonucleotides, recombinant DNA, software, and materials) are given in Supplementary Table 1.

**Cell culture**. Cells were grown in DMEM/Ham-F10 (1:1) supplemented with 10% FCS and 40 μg/ml gentamicin. Treatments with t-RA (10 μM) were performed in the presence of red phenol-free medium containing 5% charcoal-treated FCS and 40 μg/ml gentamicin.

**Generation of knock-in TFIIEβ/A150P cell line**. Human osteosarcoma epithelial U2OS cells were co-transfected with the px2-Cas9WT(GFP)-Puro plasmid (over-expressing tagged GFP-Cas9 and functional guide RNA, gRNA) and a single-stranded oligodeoxynucleotide (ssODN) corresponding to an exon 5 region of *GTF2E2* with the missense mutation c.448G>C [p.Ala150Pro]. The GFP-positive cells were sorted 24 h post-transfection, and were diluted 48 h later as 0.5 cells/well in p96 plates. Cells were grown in DMEM (1 g/l glucose) supplemented with 10% FCS and 40 μg/ml gentamicin. Clones were screened by PCR and enzymatic digestion followed by further sequencing to confirm the correct knock-in in the desired locus.

**Retrotranscription and real-time qPCR**. Total RNAs (2 μg) were reverse tran-scribed with Moloney murine leukemia virus RT (Invitrogen) using random hexanucleotides. Real-time qPCR (denaturation at 95 °C for 10 s, annealing at 65 °C for 15 s, elongation at 72 °C for 15 s) was done using the "FastStart DNA Master SYBR Green" kit and the LightCycler apparatus (Roche Diagnostics). The *RARβ2* mRNA expression represents the ratio between values obtained from treated and untreated cells normalized to 18S RNA. The primer sequences are given in Sup-plementary Table 1.

**ChIP assays**. Subconfluent cells were treated with t-RA (10 μM) and ChIP experiments were next performed with Protein G-magnetic beads (Dynabeads, Invitrogen)[59]. Non-specific controls were performed with samples incubated with magnetic beads in the absence of antibodies. Primers were designed to amplify regions located either at the proximal promoter or at exon 4 of *RARB2* (see Sup-plementary Materials)[30]. The primer sequences are given in Supplementary Table 1. PCR (denaturation at 95 °C for 10 s, annealing at 65 °C for 15 s, elongation at 72 °C for 15 s) was done using the "FastStart DNA Master SYBR Green" kit and the LightCycler apparatus (Roche Diagnostics).

**Construction and purification of rIIEs and rIIHs**. Flag-TFIIEα as well as His-TFIIEβ/WT, /A150P and /D187Y were produced in *E. coli*. Baculoviruses over-expressing Flag-XPD/WT, /R112H, /R722W, as well as XPB, p62, p52, p44, p34, cdk7, cyclin H, MAT1, and p8 were previously generated[60,61]. To purify entire rIIHs or XPD alone, Sf21 insect cells were infected with the corresponding bacu-loviruses. The whole cell extracts (WCE) were then incubated with agarose beads bound to anti-M2-Flag antibody. Recombinant TFIIH and the different forms of XPD were eluted with an epitope peptide.

**NER assays**. The single lesion (Pt-GTG) plasmid was prepared as described else-where[62]. Dual incision assay was carried out in the presence of XPG, XPF/ERCC1, XPC/hHR23B, RPA, XPA, the different rIIHs and rIIEs (when indicated)[61].

**Transcription reactions**. The AdMLP template was preincubated for 15 min at 25 °C with Pol II, TFIIA, TBP, TFIIB, TFIIF, and the different rIIEs and rIIHs[63]. RNA synthesis was initiated by the addition of NTPs (200 μM), including radi-olabelled CTP (0.15 μM). In order to study Pol II phosphorylation during tran-scription, classical run-off transcription assays were carried out in presence of cold NTPs; after SDS–PAGE, specific monoclonal antibody was used to reveal Ser5-P of Pol II.

**Abortive synthesis reactions**. The AdMLP template was preincubated with the GTFs (Pol II, TBP, TFIIB, TFIIE, and the corresponding rIIEs and rIIHs) for 15 min at 25 °C. Phosphodiester bond synthesis was then initiated by the addition of priming dinucleotides CpA (0.5 mM), MgCl₂ (6.5 mM), dATP (4 μM), and radi-olabelled CTP (1 μM). After 30 min of CpApC synthesis, the reactions were stopped in the presence of proteinase K (0.5 mg/ml).

**PIC formation assays**. Biotinylated AdMLP bound to streptavidin magnetic beads was incubated 20 min at 25 °C with Pol II, TFIIA, TBP, and the different rIIEs and rIIHs. After several washings at 50 mM NaCl, the bound fractions were resolved by SDS–PAGE followed by immunoblottings. The abundance of each

protein was assessed by immunoblot densitometry analysis (using ImageJ soft-ware). Each signal was quantified three times (mean ± s.d.) and plotted in arbitrary units (a.u.).

**Interaction assays**. Depending on the coimmunoprecipitation assay, purified Pol II, rIIEs, and rIIHs were coincubated together with specific antibody (raised against either RPB1, XPD, p62, or Flag-Tag, as indicated) bound to protein G magnetic beads. After extensive washings (at either 150 or 500 mM KCl), Western blots were performed using antibodies raised against the proteins of interest.

**Reporting summary**. Further information on research design is available in the Nature Research Reporting Summary linked to this article.

## Data availability

All relevant data supporting the key findings of this study are available within the article and its Supplementary Information files or from the corresponding author upon reasonable request. The raw data underlying all reported averages in graphs are available upon request. The source data underlying Figs. 1–6 and Supplementary Figs. 1–7 are provided as a Source Data file. A Reporting Summary for this Article is available as a Supplementary Information file.

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

## Acknowledgements

We thank Miria Stefanini, Donata Orioli, Elena Botta, Christiane Kuschal, and Kenneth H. Kraemer for the different TTD cell lines; Charlotte Saint André, Philippe Catez, and Isabelle Kolb for their technical expertise; Bernardo Reina Saint Martin for setting up the CRISPR-Cas9 technique; Donata Orioli, Kenneth H. Kraemer, Arnaud Poterszman, and Kwang-Wok Choi for fruitful discussions; Thomas Sexton for carefully reading the manuscript. We also thank the IGBMC cell culture facility. This study was supported by l'Association pour la Recherche sur le Cancer (130607082), l'Institut National du Cancer (INCA-PLBIO17-043), la Ligue contre le cancer (Equipe labellisée 2019), the PICS-CNRS (no. 6824) and the Korean National Research Foundation for international collaboration (Global Research Laboratory program).

## Author contributions

E.C., C.M.G, J.-M.E conceived and designed the experiments. E.C., C.M.G., and C.B. carried out the experiments. E.C., C.M.G., C.B., F.C., and J.-M.E. analyzed the data. E.C., F.C. and J.-M.E. contributed reagents, materials and analysis tools. E.C. and J.-M.E. wrote the paper.

## Additional information

**Competing interests:** The authors declare no competing interests.

