## [Peer Review File · Nature Communications]

Reviewers' comments:

Reviewer #1 (Remarks to the Author):

In this study, Compe et al. illuminate early events in transcription by RNA polymerase II (Pol II). Their starting point is the recent association of mutations affecting TFIIE—a component of the pre-initiation complex (PIC)—with the disease trichothiodystrophy (TTD), which is also linked to mutations in the XPD subunit of TFIIH. The authors show that TFIIE is specifically required for efficient recruitment of the CDK7 module of TFIIH into the PIC and that the TTD mutants are defective in this function. They also show a requirement for CDK7 activity (CAK), known to phosphorylate the carboxy-terminal domain (CTD) of Pol II among other targets, in dissociating itself and TFIIE α , and subsequently TFIIE β , from the PIC or initiated Pol II complex. Full TFIIE eviction occurs concomitant with, and is likely to be a prerequisite for, recruitment of the elongation factor DSIF—an exchange that the authors seem to have recapitulated *in vitro*. The paper contains many novel, interesting and potentially disease-relevant observations, and overall the strategy and experimental design are sound. There are, however, numerous instances where the authors go well beyond what the data will support: overinterpreted and potentially misinterpreted results, incorrect statements and unjustified assertions. There are missing controls, contradictory or internally inconsistent data, and effects that are too weak to support the strong causal relationships that are inferred. Although (or maybe because) I find much of the central mechanistic argument compelling, I feel strongly that these flaws would need to be corrected before I could recommend publication. My specific concerns are:

1. The authors fail to provide necessary context for their results. A role for CDK7 activity in promoting factor exchange of TFIIE and DSIF, which bind overlapping sites on Pol II, was previously demonstrated in ref. 50, which the authors cite but only to mention that TFIIE was shown in that study to be a potential CDK7 substrate. More recently, Nilson et al. recapitulated this effect, by showing that THZ1—the same CDK7 inhibitor used here—attenuated DSIF-dependent promoter-proximal pausing *in vitro*. That work (*Mol. Cell* 59: 59, 576-87, 2015) should be cited, as should Kelso et al. (*MCB* 34: 3675-88, 2014), which reported similar, reciprocal effects of a different CDK7 inhibitor on TFIIE dissociation and DSIF recruitment.
2. In the abstract, the authors state that “maintaining CAK at promoter prevents RNA synthesis” but the only way they can do this is by inhibiting its activity. (There is a CAK add-back but that's not “maintaining” anything, nor does it have a dramatic or straightforward effect.) Unless they can somehow tether CDK7 to the promoter without inhibiting it, they cannot conclude that persistent CAK occupancy *per se* is a block to transcription.
3. A very minor point, but on line 59 they cite “Recent observations” that are referenced to papers from 1996 and 1999.
4. All of the ChIP and RT-PCR analyses in Fig. 1 are done at the RAR β 2 gene. This is convenient because the gene is readily inducible, but also potentially problematic because of the direct regulation of RAR α by CDK7-mediated phosphorylation, which the Egly lab has characterized so extensively. Have they looked at another gene, where effects of CDK7 are more likely to be due exclusively to its actions on basal machinery?
5. Lines 142-6: The authors go from describing differential effects of RA on factor recruitment in wild-type versus TFIIE mutant fibroblasts to a statement of causation (“As a consequence...”) that I don't believe is justified. This is an inference from correlative data.
6. In many of the ChIP experiments (Fig. 1, Supplemental Fig. 2c), there are “missing” signals (i.e., no bars visible above x axis). I presume this is due to background (minus antibody?) signals that are higher than signals obtained with specific antibodies, but I'm afraid it does not inspire confidence. My skepticism is enhanced by a seeming lack of correlation between expression of the RAR β 2 gene (by RT-PCR analysis, Fig. 1d) and occupancy by RPB1, Ser5-P, Ser2-P and SPT5 in the downstream regions of the gene (Fig. 1f).
7. In Supplemental Fig 2d, a necessary control for total RAR α recovered in the IP is missing; only phosphorylated RAR α is shown.
8. In Fig. 1e, f, the authors measure effects of a TFIIE mutation on CTD phosphorylations (Ser5-P and Ser2-P) by ChIP, but at many positions along the gene these effects are difficult to interpret because of (sometimes larger) effects on total Pol II occupancy. It would be more rigorous to present the ratio of phospho- to total signals in addition to the absolute levels.
9. The lack of effect of TFIIE mutations on NER (Fig. 2c) and the contrast with XPD mutants (2c)

are essentially controls that could be moved to the supplement.

10. I am curious as to why the A150P mutant has so much stronger effects *in vitro* (lines 196-8) when their effects *in vivo* appeared to be similar (Fig. 1). Do the authors have any explanation for this discrepancy?

11. Line 215: In describing immunoblot analysis of immobilized template assays, the authors should avoid the subjective language used to distinguish between effects on XPD (“barely recruited”) and CDK7 (“nearly absent”). They don’t actually say how they quantify the immunoblot signals to generate the histogram in Fig. 3a, but looking at the blots themselves, I don’t think any quantitative distinction is possible; differences could be due to different antibodies with different thresholds of detection. Based on this experiment, all one can really say is that both signals are reduced, and that the effect on CDK7 might be more severe—a point that data presented later seem to support.

12. Line 218: Similarly, I would not say binding of CAK is “disrupted” in the mutants, based on the <50% reduction in western blot signals, which are still way above background and reduced roughly proportionally with those of the TFIIE signals.

13. Similarly, the effects on co-immunoprecipitation (Fig. 3b) are quite minimal (and not quantified).

14. Lines 223-231: I found this passage—which describes a set of observations that form the core of their mechanistic model—confusing and hard to follow. This is in part due to the unorthodox use of verbs such as “target” and “integrate” in place of (I think) “bind” and “recruit.” But aside from that, it is hard to discern what I think are the main points, namely, that TFIIE β but not TFIIE α can bind TFIIH in solution (Fig. 3d), whereas both subunits are necessary to recruit the CAK module—and to enhance binding of core TFIIH—to AdMLP (Fig. 3c). Is this correct? Am I missing something?

15. Line 234-5, referring to Fig. 3b: Again the word “disrupted” (albeit with the qualifier “partially”) is used to describe a modest effect (on TFIIE-IIH co-IP) that is not quantified. If anything, the D187Y mutation seems to be making the TFIIH- TFIIE β interaction stronger, *i.e.*, more salt-resistant (compare lane 6 to lane 2).

16. Lines 240-241: I see no evidence to support the assertion that the TFIIE or XPD mutations affected (or “disrupted”) “accurate positioning of the CAK at the AdMLP”. This wording implies a structural insight that the data cannot provide.

17. Lines 249-50: A minor point, but it’s not accurate to say that Pol II phosphorylation is essential for transcription initiation (although it is essential for viability). As an example, Ser5-P, the major phosphorylation ascribed to TFIIH, can be bypassed in yeast, *e.g.* by RPB1 fusion to the capping enzyme (Schwer and Shuman, *Mol. Cell* 43:311-8, 2011), meaning it cannot be strictly essential for initiation.

18. Lines 255-7, referring to Fig. 4a: ATP-Y-S is NOT non-hydrolyzable, as stated; many kinases (including many CDKs) can use it to thiophosphorylate their substrates, albeit typically less efficiently than they use natural ATP. In fact, there is a band (or rather half a band, partially obscured by what looks like a blotting artifact) detected with the Ser5-P antibody in lane 4. This treatment might be reducing, not abolishing CDK7 activity towards Ser5, but I’m not aware of any data regarding the ability of commercial anti-Ser5-P antibodies to recognize thiophosphorylated Ser5, so even that cannot be concluded with any certainty. I would tend to interpret the effects of this analog on dissociation of CAK and TFIIH subunits from the AdMLP as likely due to slower turnover of the enzyme.

19. Lines 259-63, referring to Fig. 4c: The effects of the CDK7 inhibitor THZ1 on Ser5-P are quite modest, even at the (very) high concentration of 10 μ M. Although this does seem to correlate with increased retention of TFIIE α and maybe cyclin H, the effects are not dramatic. Have the authors tried a pre-incubation with this compound, which was used by Nilson *et al.* (cited above), and rationalized on the basis of THZ1’s covalent (and thus slow) mechanism of inhibition?

20. In Fig. 4d-f, there is an apparent effect of CDK7 activity on electrophoretic mobility of TFIIE α , which the authors do not comment on, namely the band becomes a doublet with a slower-migrating component. This might be interesting; TFIIE α has been reported to be a TFIIH/CDK7 substrate (refs. 9, 50). It also raises the potential concern that the apparent loss of signal is due in part to the splitting into two bands. A phosphatase treatment prior to SDS-PAGE might help to resolve this issue.

21. Lines 278-80 and elsewhere: Here the authors conclude correctly that, based on data obtained with the XPB inhibitor triptolide and with an XPB-inactivating mutation, “promoter opening was not a prerequisite for Pol II phosphorylation” and steps downstream. But elsewhere they say,

inaccurately, that the data show CAK action preceding opening (Abstract, line 25, lines 266-7; Discussion, line 368), implying a temporal sequence that cannot be assumed from the fact that the two processes can be uncoupled.

22. Line 283: The subheading makes no sense; what is meant by “conditions”? (This verb is also used, confusingly, elsewhere in the manuscript.)

23. Lines 284-314, describing Fig. 5. To address whether CAK release is important for transcription, the authors use a very different experimental set-up, in which that release is first allowed to occur, transcription complexes are re-isolated, and “free” CAK is added back. As noted above (point 2), this is not “maintaining” CAK at the promoter (which I cannot think of a straightforward way to do, short of covalently tethering it to another TFIID subunit). The mild inhibitory effects of this treatment on transcription are difficult or impossible to interpret. Indeed, the effects of THZ1 on transcription—without an apparent effect on Ser5-P and in a system from which CAK has been depleted—are even greater. The authors note this effect and ascribe it to residual promoter-bound CAK, but don’t address the question of what it might be doing to promote transcription, or what it might be phosphorylating. The additive effects of CAK and THZ1 addition—on both CAK recruitment and transcription—also lack an explanation. Finally, the correlation between eviction of TFIIE β and TFIID, and recruitment of SPT5 from a crude extract (Fig. 5c), is intriguing, but was obtained under quite different experimental conditions, and raises some questions, for example: What happens when the reactions are supplemented with ATP only, without the other 3 NTPs? In any case, the statement concluding this section, that “CAK release is a prerequisite for RNA synthesis” is far stronger than what the data will support, and should be deleted or toned down.

24. Line 316: The subheading of this, the final section of Results, is simply not supported by any of the data shown in Fig. 6. I am mostly convinced that the TTD mutations in TFIIE β and XPD are impeding CAK recruitment to some degree (“coming”) but not its dissociation upon ATP addition (“going”).

Reviewer #2 (Remarks to the Author):

Review of Nature Comm. ms 18-29809, by Compe et al.

In this paper the authors explore the involvement of TFIIE in PIC assembly, particularly in the context of the interplay of TFIIE and TFIID. Important tools for this work are known mutations in TFIIE subunits that affect RNA synthesis. The results reported here refine our ideas on the role of TFIIE on the loading of TFIID, and of the CAK subcomplex in particular, into the PIC. Without recapping all of the findings, I think the most important contribution here is the demonstration that CAK and TFIIE alpha apparently exit the PIC upon ATP addition and CTD modification, prior to template melting. The authors maintain that retention of CAK in the complex impedes RNA synthesis.

The full function of TFIIE in the pol II PIC and its relationship with the loading and function of TFIID remain incompletely understood. As I just noted, there are some potentially important advances in this work, but I also felt that there were many perplexing aspects and unanswered questions raised that the authors should have addressed. These are noted below in connection with the relevant figures.

Fig. 1- The results here are intriguing but they are not fully consistent across all of the assays. Consider panels D and E, and compare the effect of the E mutant on pol II levels (RPB1) at the promoter versus pol II levels at exon 4, particularly at 8 hr. At 8 hr, there is essentially no drop in gene expression or in pol II levels at the promoter because of the E mutant. However, there is far less pol II at exon 4 in the case of the mutant, which seems at odds with the gene expression result. The authors don’t address this point. Importantly, these points are a hint that the E mutant has post-initiation effects, which is significant when considering some of the other results in this paper.

Fig. 2- The in vitro assay (as in panel E) is not well described- the reference is from 1991. Are assembly reactions still done for 15 min? If so, did the authors check to see if assembly was essentially complete in 15 min for both wt and mutant IIE? That is, is the mutant IIE defect related to failure in assembly, in function of the assembled complexes, or both? My impression is that such classic assays essentially report on a single round of transcription. This is important when comparing the effect of the E mutants, particularly D187Y, in the run-off versus the abortive

assay. D187Y IIE works almost as well as wt in the abortive assay but much less well in run-off. Abortive synthesis of CAC is clearly multiround. Does this mean that complexes that contain D187Y IIE initiate well but fail to clear the promoter, perhaps because XPB cannot effectively support advance out to $\sim +10$? Is a failure in transcription early, but post-initiation, somehow linked to the anomaly noted above with Fig. 1?

Fig. 3- These results reinforce the well-established idea that IIE is important for the effective recruitment of IIH, and they advance our understanding by further discriminating between the individual roles of IIE alpha and beta. However, the functional points of several of the Fig. 3 results are not clear. For example, in panel A we see that without any IIE, the core of IIH is still recruited, at perhaps 40% of its optimal level. Do those complexes make any RNA (abortive or run-off)? When either mutant IIE is used, the core of IIH is recruited as well as when wt IIE is used, but we know from Fig. 2 that these complexes are deficient to varying degrees in RNA synthesis assays. A potential problem which affects all experiments like those in this figure is the relationship between GTF assembly as assayed by western and the level of functional PICs assembled. Do the authors have any insight into this point- that is, what fraction of the apparent PICs actually make RNA? This is relevant in considering some of the other results. In particular, it will be shown that CAK is lost upon ATP addition and CAK retention can affect transcriptional competence of the complex. Is RNA synthesis the same (abortive and run-off) when complexes are assembled with core IIH only, as compared to complexes that have lost CAK as a result of ATP addition (and thus also have phosphorylated CTDs)?

Fig. 4- Since the exact experimental procedure is not fully described, it is not clear if the assays with triptolide are meaningful. Since TPL is a covalent inhibitor, it takes many minutes for full inhibition of XPB to occur, while promoter opening is, as far as I know, very rapid. Thus, to conclude that promoter opening did not happen I would want to see preincubation with TPL for 10 min, only then followed by addition of ATP. Fortunately for the authors, I think they are on solid ground with the XPB mutants, so the point about CAK loss prior to promoter opening seems well-supported. Without additional tests I would not include the TPL results.

Fig. 5- Panel A indicates that a substantial fraction of IIE, XPB and pol II remain on the template after chase with all NTPs. Pol II retention can occur the end of template DNAs even after "run-off", depending on the nature of the template end and the reaction conditions. However, based on the structure of the PIC and the initiating complex, it does not seem reasonable that any IIE or IIH should remain on the DNA after promoter clearance. This again raises the question of whether a substantial fraction, perhaps the majority, of the GTFs retained on the template are not in functional transcription complexes. Does this mean that most of the signal observed in the western comes from inactive complexes, when the complexes are assembled from purified components? This worry presumably dictated the panel C experiment with nuclear extract addition, where as expected XPB and (mostly) IIE are lost on chase- but pol II is still retained.

I think there is an important control missing from panel A. The complexes were incubated with ATP for 15 min, after which a 309 nt run-off RNA could be made. I found this surprising, since there is an extensive earlier literature showing that pol II PICs have a very limited lifespan after ATP incubation (see: JBC 271, p. 7245; EMBO J 16, p. 7468; JBC 287, p. 961). This suggests a clear question- if the complexes in Fig. 5 had been incubated for 15 min without ATP and then chased, how much run-off RNA would have been made? Is there a substantial loss of RNA synthesis upon extended incubation with ATP in this system? Regardless of the answer, it is important to place the current work in the context of these well-established earlier findings.

The THZ1 result (panel B, lanes 4 and 5) is very hard to explain. The authors suggest that this may have resulted from residual CAK bound to the promoter, probably because of the synergy observed with added CAK and THZ1, but at least in Fig. 5 there is no evidence of any CAK in the complexes when additional CAK is not added (lanes 4, 5 of panel B).

The authors interpret the Fig. 5B results to show that retention of CAK impedes RNA synthesis. One could imagine that CAK release is important in allowing XPB to effectively drive DNA into the PIC within the overall IIH complex, so blocking CAK release could inhibit open complex formation, consistent with CAK release before template opening. But, the add-back experiment is not a direct test of that idea. The bead-bound PICs had seen ATP and thus the templates were melted and the CTDs were modified before washing and re-addition of CAK. I am not sure what mechanism the authors have in mind for CAK inhibition in this case.

Overall, there are a number of interesting findings in this paper but as I noted above, there are

significant ambiguities in the results in several places. In particular, there are important points where considerably more could have been learned if changes in transcription complex assembly/composition had been linked to changes in ability to make RNA. Finally, I note an important typo- on p. 3 in the last paragraph, the list of the components of TFIID-Core does not include XPD.

Reviewer#1

We thank reviewer for finding that “the paper contains many novel, interesting and potentially disease-relevant observations and overall the strategy and experimental design are sound”. According to her/his comments, new experiments have been performed, as outlined below, and several modifications have been done to clarify the manuscript.

1. The authors fail to provide necessary context for their results. A role for CDK7 activity in promoting factor exchange of TFIIE and DSIF, which bind overlapping sites on Pol II, was previously demonstrated in ref. 50, which the authors cite but only to mention that TFIIE was shown in that study to be a potential CDK7 substrate. More recently, Nilson et al. recapitulated this effect, by showing that THZ1—the same CDK7 inhibitor used here—attenuated DSIF-dependent promoter-proximal pausing in vitro. That work (Mol. Cell 59: 59, 576-87, 2015) should be cited, as should Kelso et al. (MCB 34: 3675-88, 2014), which reported similar, reciprocal effects of a different CDK7 inhibitor on TFIIE dissociation and DSIF recruitment.

As suggested by the reviewer, we provide additional references (page 16): “Interestingly, it has been recently described reductions in DSIF and increases in TFIIE near promoters as a result of CDK7 inhibition (Kelso et al., MCB, 2014; Nilson et al., Mol Cell, 2015).”

2. In the abstract, the authors state that “maintaining CAK at promoter prevents RNA synthesis” but the only way they can do this is by inhibiting its activity. (There is a CAK add-back but that’s not “maintaining” anything, nor does it have a dramatic or straightforward effect.) Unless they can somehow tether CDK7 to the promoter without inhibiting it, they cannot conclude that persistent CAK occupancy per se is a block to transcription.

Indeed, we cannot conclude that persistent CAK occupancy is a block to transcription. Accordingly, modifications were done:

In the abstract: “Strikingly, maintaining CAK at promoter prevents RNA synthesis” has been removed. In the discussion (page 16): “Moreover, it even seems that the CAK inhibition affects the proper RNA synthesis (Fig. 5b)”

3. A very minor point, but on line 59 they cite “Recent observations” that are referenced to papers from 1996 and 1999.

“Recent” has been removed (page 3).

4. All of the CHIP and RT-PCR analyses in Fig. 1 are done at the RAR β 2 gene. This is convenient because the gene is readily inducible, but also potentially problematic because of the direct regulation of RAR α by CDK7-mediated phosphorylation, which the Egly lab has characterized so extensively. Have they looked at another gene, where effects of CDK7 are more likely to be due exclusively to its actions on basal machinery?

The expression of housekeeping genes such as GAPDH was similar in WT and TFIIE mutated cells (see panel a, Annex Reviewer#1). To study *in vivo* transcription failures resulting from TFIIE and TFIIH mutations, we and others usually investigated the behavior of inducible genes such as RAR β 2. Our results suggest that the disruption of the RAR β 2 mRNA synthesis result neither from the nuclear receptor phosphorylation defect, nor from RAR α recruitment on RAR β 2 promoter (Supplemental Fig. 2c-d). It seems that the RAR β 2 expression failure was due to defects during PIC assembly.

5. Lines 142-6: The authors go from describing differential effects of RA on factor recruitment in wild-type versus TFIIE mutant fibroblasts to a statement of causation (“As a consequence...”) that I don’t believe is justified. This is an inference from correlative data.

The sentence has been changed (page 7):

“Furthermore, an improper Ser5-P of Pol II has been observed at the RAR β 2 promoter”).

6. In many of the ChIP experiments (Fig. 1, Supplemental Fig. 2c), there are “missing” signals (i.e., no bars visible above x axis). I presume this is due to background (minus antibody?) signals that are higher than signals obtained with specific antibodies, but I’m afraid it does not inspire confidence. My skepticism is enhanced by a seeming lack of correlation between expression of the *RARβ2* gene (by RT-PCR analysis, Fig. 1d) and occupancy by RPB1, Ser5-P, Ser2-P and SPT5 in the downstream regions of the gene (Fig. 1f).

The absence of signals for TFIIB, TFIIE and (to a lower extent) TFIIF/CDK7 at exon 4 (which is located at 140kb from the transcription start site) reflects the fact that these general transcription factors do not follow elongating Pol II (Spangler et al., PNAS, 2001; Le May et al., Mol Cell, 2010, Larochelle et al., NS&MB, 2012). Significant signals for these factors were however observed at the proximal promoter of *RARβ2*, which strengthens the reliability of our ChIP experiments.

In order to evaluate the consequences of TFIIE mutations on elongation, we performed ChIP experiments on phosphorylated Pol II at exon 4. In the KI-IIE/A150P (TFIIE mutated) cells, we first observed a significant decrease of Ser5-P at exon 4, compared to wild-type (Fig. 1f1 and 1f5). Moreover, mutation in TFIIE also resulted in a lower presence of Ser2-P. Hence, the impaired recruitment of TFIIE in KI-IIEβ/A150P at the *RARβ2* promoter, and the gene expression defect, might be likely due to some failure in Pol II phosphorylation, a point that will be dissected in the next experiments.

The referee is concerned about “the lack of correlation between expression of the *RARβ2* gene and occupancy by RPB1, Ser5-P, Ser2-P and SPT5 at exon 4”. We would like to mention that both *RARβ2* expression as well as the recruitment of the transcriptional machinery peaked together at the promoter 6h post-treatment in WT cells (Fig. 1d and 1e1-e7), as previously observed (Singh et al., AJHG, 2015). A correlation between RPB1, Ser5-P, Ser2-P and SPT5 4 and *RARβ2* expression also seemed to occur at the promoter in KI-IIEβ/A150P cells. However, the “lack of correlation” at exon 4 could find explanations by considering the following points: (i) the *RARβ2* mRNA has a relatively short half-life; (ii) the transcriptional activation of the *RARβ2* gene is a cyclical process (see panel b, Annex Reviewer#1) (Le May et al., Mol Cell, 2010; Singh, AJHG, 2015); (iii) In WT cells, elongating Pol II (and factors such as SPT5) might be much more present at exon 4 during intermediate times (between 6 and 8h) of t-RA treatment; Pol II accumulation might result from both the first and the subsequent cycle. This might differ in KI-IIEβ/A150P cells, in which *RARβ2* expression peaked at 8h and the subsequent transcription cycles might start later.

This point has been addressed in the manuscript (page 7).

“The fact that signals for phosphorylated elongating Pol II have been observed at 8h while the *RARβ2* mRNA tend to decrease might be due to the cyclic profile of the *RARβ2* gene expression (Le May et al., Mol Cell, 2010) and the very distal position of exon 4 from the transcription start site (~140kb).”

7. In Supplemental Fig 2d, a necessary control for total *RARα* recovered in the IP is missing; only phosphorylated *RARα* is shown.

Similar amounts of *RARα* have been found in WT and TFIIE mutated whole cell extracts after immunoblot analysis using antibodies raised against *RARα* (β-actin has been used as control, Supplementary Fig. 2c). Moreover, upon t-RA treatment, similar amounts of *RARα* were targeting the *RARβ2* promoter (Supplementary Fig 2d and 2c). Altogether, our observations suggest that i) TFIIE mutations did not affect the amount of *RARα* and its recruitment at the promoter of target gene and ii) *RARα* phosphorylation was not affected by TFIIE mutations.

8. In Fig.1e, f, the authors measure effects of a TFIIE mutation on CTD phosphorylation (Ser5-P and Ser2-P) by ChIP, but at many positions along the gene these effects are difficult to interpret because of (sometimes larger) effects on total Pol II occupancy. It would be more rigorous to present the ratio of phospho- to total signals in addition to the absolute levels.

The referee is right when pointing out that these effects are difficult to interpret. In addition, it’s never easy to compare data resulting from the use of different antibodies as well as from different set of

cells (even though having similar genetic background). Nonetheless, although Ser5-P was similar at the RAR β 2 promoter on both cell lines, it was highly decreased at exon 4 in KI-IIE β /A150P cells (Fig. 1e4 and 1f4); such Pol II phosphorylation defect was even more visible when considering Ser2-P (Fig. 1f6). However, and according to the referee's suggestion, we analyzed the Ser5-P/Pol II and Ser2-P/Pol II ratios. At the promoter, the ratio Ser5-P/Pol II was nearly similar in WT and KI-IIE β /A150P cells (at 6h, 1.4 and 1.6, and at 8h, 1.25 and 1.4, respectively). While the Ser2-P/Pol II ratios were comparable in WT and TFIIE mutated cells at 6h, differences have been obtained at 8h, reflecting the defect of Pol II phosphorylation once TFIIE was mutated. At exon 4, some ratios could not be determined due to the absence of detected signals (as previously discussed, point 6), especially at 6h in KI-IIE β /A150P cells. Nonetheless, the Ser5-P/Pol II and Ser2-P/Pol II ratios seemed to be quite similar in both cells at 8h. Taking together, such ratios in addition to our previous analysis, provide similar conclusions and show that the defect observed at the formation of the PIC has consequences during the elongation step.

9. The lack of effect of TFIIE mutations on NER (Fig. 2c) and the contrast with XPD mutants (2c) are essentially controls that could be moved to the supplement.

These results have been moved to the supplement (Supplementary Fig.3).

10. I am curious as to why the A150P mutant has so much stronger effects in vitro (lines 196-8) when their effects in vivo appeared to be similar (Fig. 1). Do the authors have any explanation for this discrepancy?

Our reconstituted in vitro assays (in which we only used the basal transcription machinery) allowed us to study the transcriptional consequences of TFIIE/A150P and /D187Y (Fig. 2c and Fig. 2e). In a cellular context, such comparison between TFIIE/A150P and /D187Y requires taking into account additional parameters: (i) the genetic background differs between the mutated fibroblasts. (ii) the differences of failures resulting from TFIIE mutations might be attenuated by other transcription factors recruited at the RAR β 2 promoter, (iii) the induction of the RAR β 2 gene (upon t-RA treatment) might differ between the two mutated cell lines. Indeed, the RAR β 2 expression has been analyzed after 6 and 8h of t-RA treatment, i.e the peak of induction of this gene in normal cells (Le May et al., Mol. Cell, 2010; Singh et al., AJHG, 2015). However, we cannot exclude that subtle differences might be observed during accurate kinetic analysis.

11. Line 215: In describing immunoblot analysis of immobilized template assays, the authors should avoid the subjective language used to distinguish between effects on XPD ("barely recruited") and CDK7 ("nearly absent"). They don't actually say how they quantify the immunoblot signals to generate the histogram in Fig. 3a, but looking at the blots themselves, I don't think any quantitative distinction is possible; differences could be due to different antibodies with different thresholds of detection. Based on this experiment, all one can really say is that both signals are reduced, and that the effect on CDK7 might be more severe—a point that data presented later seem to support.

The abundance of each protein was assessed by immunoblot densitometry analysis (using ImageJ software). The signals were quantified (mean \pm s.d.) and plotted in arbitrary units (au). This is now included in the manuscript (Material and methods, "PIC formation assays" section).

Furthermore, the sentence has been modified (page 9): "The signals corresponding to the Core-TFIIEH (XPB) and the XPD bridging component of TFIIEH were however reduced, such effect being more severe for CAK (CDK7)."

12. Line 218: Similarly, I would not say binding of CAK is "disrupted" in the mutants, based on the <50% reduction in western blot signals, which are still way above background and reduced roughly proportionally with those of the TFIIE signals.

This sentence has been changed (page 9): "Strikingly, rIIE α β /A150P and /D187Y reduced the binding of the CAK (CDK7) [...]"

13. Similarly, the effects on co-immunoprecipitation (Fig. 3b) are quite minimal (and not quantified).

The signals were quantified (mean±s.d.) (see new Fig3b). Graph shows in arbitrary units (au) the ratio IIE α /IIE β for each rIIE after coimmunoprecipitation with rIIH.

14. Lines 223-231: I found this passage—which describes a set of observations that form the core of their mechanistic model—confusing and hard to follow. This is in part due to the unorthodox use of verbs such as “target” and “integrate” in place of (I think) “bind” and “recruit.” But aside from that, it is hard to discern what I think are the main points, namely, that TFIIE β but not TFIIE α can bind TFIIH in solution (Fig. 3d), whereas both subunits are necessary to recruit the CAK module—and to enhance binding of core TFIIH—to AdMLP (Fig. 3c). Is this correct? Am I missing something?

This part has been modified (page 9): “In absence of rIIE β /WT, rIIE α /WT was unable to bind the AdMLP/Pol II/TBP/TFIIA/TFIIB/TFIIF complex (Fig. 3c, lane 2). Strikingly, while TFIIH interacted with rIIE β /WT rather than rIIE α /WT in solution (Fig. 3d, lanes 2 and 4), both TFIIE subunits were required to fully recruit the entire TFIIH complex, including CAK, at the AdMLP (Fig. 3c, lane 7).”

15. Line 234-5, referring to Fig. 3b: Again, the word “disrupted” (albeit with the qualifier “partially”) is used to describe a modest effect (on TFIIE-IIH co-IP) that is not quantified. If anything, the D187Y mutation seems to be making the TFIIH- TFIIE β interaction stronger, i.e., more salt-resistant (compare lane 6 to lane 2).

The sentence has been modified (page 10): “The coimmunoprecipitation between TFIIE and TFIIH components were slightly reduced in the presence of rIIE $\alpha\beta$ /A150P or rII $\alpha\beta$ /D187Y variants”.

Undoubtedly, structural analyses might be helpful to accurately determine the consequences of the A150P and D187Y substitutions in the TFIIE stability as well as in the TFIIE-TFIIH partnership.

16. Lines 240-241: I see no evidence to support the assertion that the TFIIE or XPD mutations affected (or “disrupted”) “accurate positioning of the CAK at the AdMLP”. This wording implies a structural insight that the data cannot provide.

This confusing sentence has been modified (page 10): “As a consequence, the recruitment of the CAK at the AdMLP was reduced, as observed for XPD/R722W (Fig. 3g, compare lanes 5 to 3).”

17. Lines 249-50: A minor point, but it’s not accurate to say that Pol II phosphorylation is essential for transcription initiation (although it is essential for viability). As an example, Ser5-P, the major phosphorylation ascribed to TFIIH, can be bypassed in yeast, e.g. by RPB1 fusion to the capping enzyme (Schwer and Shuman, Mol. Cell 43:311-8, 2011), meaning it cannot be strictly essential for initiation.

This sentence has been changed (page 10): “We next investigated how TFIIE and TFIIH cooperate to phosphorylate Pol II and to open DNA during transcription.”

18. Lines 255-7, referring to Fig. 4a: ATP- γ S is NOT non-hydrolyzable, as stated; many kinases (including many CDKs) can use it to thiophosphorylate their substrates, albeit typically less efficiently than they use natural ATP. In fact, there is a band (or rather half a band, partially obscured by what looks like a blotting artifact) detected with the Ser5-P antibody in lane 4. This treatment might be reducing, not abolishing CDK7 activity towards Ser5, but I’m not aware of any data regarding the ability of commercial anti-Ser5-P antibodies to recognize thiophosphorylated Ser5, so even that cannot be concluded with any certainty. I would tend to interpret the effects of this analog on dissociation of CAK and TFIIH subunits from the AdMLP as likely due to slower turnover of the enzyme.

This sentence has been modified (page 10): “In presence of ATP- γ S, whose hydrolysis is very low compared to ATP, neither Pol II phosphorylation nor the release of the CAK and TFIIE α occurred.”

19. Lines 259-63, referring to Fig. 4c: The effects of the CDK7 inhibitor THZ1 on Ser5-P are quite modest, even at the (very) high concentration of 10 μ M. Although this does seem to correlate with increased retention of TFIIE α and maybe cyclin H, the effects are not dramatic. Have the authors

tried a pre-incubation with this compound, which was used by Nilson et al. (cited above), and rationalized on the basis of THZ1's covalent (and thus slow) mechanism of inhibition?

TFIIH (Fig. 4c) as well as CAK (Fig. 5b) have been pre-incubated (10min at 25°C) with THZ1 before to be added in the in vitro PIC assays.

20. In Fig. 4d-f, there is an apparent effect of CDK7 activity on electrophoretic mobility of TFIIEx, which the authors do not comment on, namely the band becomes a doublet with a slower-migrating component. This might be interesting; TFIIEx has been reported to be a TFIIH/CDK7 substrate (refs. 9, 50). It also raises the potential concern that the apparent loss of signal is due in part to the splitting into two bands. A phosphatase treatment prior to SDS-PAGE might help to resolve this issue.

Indeed, a doublet for TFIIEx appeared in presence of ATP, which might be due to the phosphorylation of TFIIEx. As suggested, upon phosphatase treatment, only one band was observed, suggesting that TFIIEx was phosphorylated (see panel c, lane 4, Annex Reviewer#1). To control the efficiency of the phosphatase treatment, a Ser5-P antibody has been used, showing no more phosphorylation of Pol II upon CIAP treatment (lane 4). Note that the electrophoretic mobility of IIE α after CIAP treatment (lane 4) was different than that observed without phosphatase treatment (lane 2); this is due to the fact that we used a significant amount of CIAP (detectable on the membrane upon Ponceau S), which migrates (~70kDa) to a level similar to that of IIE α (56kDa).

Substantial investigations should be undertaken to further understand the role of the TFIIEx phosphorylation during transcription. According to the referee's comment, this point has been addressed in the discussion (page 16): "Whether or not the eviction of TFIIEx might imply post translational modifications remains to be determined. In this regard, it should be pointed out that a doublet for TFIIEx has been observed in presence of ATP (in particular Fig 4, 5 and 6). This might result from the phosphorylation of TFIIEx, likely by CDK7, as previously observed (Ohkuma et al., Nature, 1994; Larochelle et al., NS&MB, 2012). Although such apparent TFIIEx phosphorylation is of interest, substantial investigations should be undertaken to identify the phosphorylation(s) sites(s) and to determine the function(s) of such modification(s) during transcription."

21. Lines 278-80 and elsewhere: Here the authors conclude correctly that, based on data obtained with the XPB inhibitor triptolide and with an XPB-inactivating mutation, "promoter opening was not a prerequisite for Pol II phosphorylation" and steps downstream. But elsewhere they say, inaccurately, that the data show CAK action preceding opening (Abstract, line 25, lines 266-7; Discussion, line 368), implying a temporal sequence that cannot be assumed from the fact that the two processes can be uncoupled.

According to the reviewer's comment, changes have been made in the manuscript according to the original version.

(Abstract, page 2): "In addition, we show that while RNA polymerase II phosphorylation and DNA opening occur, CAK and TFIIEx are released from the promoter."

(Results, page 11): "We next examined whether the promoter opening was required for the Pol II phosphorylation and the release of TFIIEx and the CAK."

(Discussion, page 16): "Our results show that the Pol II phosphorylation and the release of TFIIEx and the CAK do not require DNA opening (Fig. 4d-f)."

22. Line 283: The subheading makes no sense; what is meant by "conditions"? (This verb is also used, confusingly, elsewhere in the manuscript.)

The verb "condition" has been removed from the manuscript.

Page 11: "The CAK is released during RNA synthesis".

Page 15: "we demonstrate that TFIIEx promotes the incorporation of [...]".

Page 38: "The CAK is released during RNA synthesis"

23. Lines 284-314, describing Fig. 5. To address whether CAK release is important for transcription, the authors use a very different experimental set-up, in which that release is first allowed to occur,

transcription complexes are re-isolated, and “free” CAK is added back. As noted above (point 2), this is not “maintaining” CAK at the promoter (which I cannot think of a straightforward way to do, short of covalently tethering it to another TFIIF subunit). The mild inhibitory effects of this treatment on transcription are difficult or impossible to interpret. Indeed, the effects of THZ1 on transcription—without an apparent effect on Ser5-P and in a system from which CAK has been depleted—are even greater. The authors note this effect and ascribe it to residual promoter-bound CAK, but don’t address the question of what it might be doing to promote transcription, or what it might be phosphorylating. The additive effects of CAK and THZ1 addition—on both CAK recruitment and transcription—also lack an explanation.

The Ser5-P signal was not increased after CAK supplementation (Fig. 5b, lanes 2-3), demonstrating that Pol II bound to the promoter was fully phosphorylated following ATP pre-incubation and addition of NTP (lane 1).

The CAK supplementation led to a significant increase of its recruitment, and a slight decrease of RNA synthesis (lanes 2-3).

The addition of THZ1 alone (lanes 4-5) led to a partial decrease of RNA synthesis (lanes 4-5), which might be due to the inhibition of residual CAK and to side effects (intercalating and/or interfering with DNA template) resulting from the concentrations of THZ1 used (1-10 μ M) and the absence of consistent amount of CDK7.

Addition of both CAK and THZ1 resulted in a higher (additive) transcription reduction (lanes 6-7). In this later case, THZ1-inactivated CAK was highly recruited at the DNA template, which might be related to the fact that TFIIE α (which has been added Fig. 5b) was maintained within the PIC (compare lane 1 and lanes 6-7). Strikingly, similar observations have been obtained upon addition of CAK/CDK7as (a CAK containing a CDK7 analog-sensitive mutant) (Larochelle et al., NS&MB, 2006; Larochelle et al., Mol Cell, 2007) and 3-MB-PP1 (an ATP analog competitive inhibitor); the recruitment of CAK/CDK7as was much higher likely due to its anchoring at the promoter, TFIIE α being not released (new Supplemental Fig. 6a).

As mentioned by the reviewer, we cannot conclude that “maintaining” CAK per se prevents transcription. However, supplementation of inactive CAK maintained a preinitiation complex containing an already phosphorylated Pol II, which prevented the subsequent steps of transcription. This part of the results has been modified (page 12) and supplemented Figure (Supplemental Fig. 6a) has been added.

“Addition of THZ1 led to a partial decrease of RNA synthesis (lanes 4-5), which may be due to the inhibition of residual CAK bound to the promoter, to side effects resulting from the concentrations of THZ1 used (1-10 μ M) and/or the absence of consistent amount of CDK7. Strikingly, addition of THZ1-inactivated CAK resulted in a higher transcription reduction (lanes 6-7). While such supplementation did not alter Pol II phosphorylation, a massive recruitment of THZ1-inactivated CAK has been observed, which might be related to the fact that TFIIE α was maintained within the PIC (compare lane 1 and lanes 6-7). Similar observations have been obtained upon addition of CAK/CDK7as (a CAK containing a CDK7 analog-sensitive mutant) (Larochelle et al., 2006; Larochelle et al., 2007) and 3-MB-PP1 (an ATP analog competitive inhibitor); the recruitment of CAK/CDK7as was much higher likely due to its anchoring at the promoter, TFIIE α being not released (Supplemental Fig. 6a). It is worthwhile to notice that supplementation with rIIE α /WT led to a slight increase of its recruitment at the promoter, without modifying the behavior of the CAK and the phosphorylation of Pol II (Supplementary Fig. 6b).”

Finally, the correlation between eviction of TFIIE β and TFIIF, and recruitment of SPT5 from a crude extract (Fig. 5c), is intriguing, but was obtained under quite different experimental conditions, and raises some questions, for example: What happens when the reactions are supplemented with ATP only, without the other 3 NTPs?

Accordingly, new immobilized *in vitro* assays were supplemented with whole cell extracts (WCE, isolated from HeLa cells) in presence of ATPs (see panel d, Annex Reviewer#1). We observed partial removal of the Core-TFIIF and TFIIE β (lane 3). Interestingly, the elongation factor DSIF (SPT5) was not recruited when compared to what obtained in presence of NTP (lane 4).

In any case, the statement concluding this section, that “CAK release is a prerequisite for RNA synthesis” is far stronger than what the data will support, and should be deleted or toned down.

The statement concluding this section has been changed (page 13): “Taking together, these results strongly suggested that TFIIE α and the CAK, followed by the Core-TFIIH and TFIIE β , are sequentially released from promoter while elongation factors take place to pursue transcription.”

24. Line 316: The subheading of this, the final section of Results, is simply not supported by any of the data shown in Fig. 6. I am mostly convinced that the TTD mutations in TFIIE β and XPD are impeding CAK recruitment to some degree (“coming”) but not its dissociation upon ATP addition (“going”).

According to the reviewer, this subheading has been changed (page 13): “By preventing CAK recruitment, TTD mutations affect Pol II phosphorylation.”

Legends - Annex Reviewer#1

panel a: the amount of GAPDH mRNA has been measured by RT-PCR in WT (KI-IIE β /WT, dark boxes) and TFIIIE mutated cells (KI-IIE β /A150P, open boxes) cultured in basal condition. The results are representative of two independent experiments performed in triplicates as indicated by standard deviation

panel b: relative RAR β 2 gene expression have been measured by RT-PCR after different times of t-RA treatment. The mRNA levels were normalized to the 18S RNA amount. The results are presented as n-fold induction relative to non-treated cells.

panel c: streptavidin magnetic beads were incubated (when indicated, +) with biotinylated AdMLP, Pol II, TFIIA,-B,-D (TBP), rIIE $\alpha\beta$ /WT,-F and TFIIH/WT, in presence of ATP (200 μ M). After 15 min of incubation, phosphatase (calf intestinal alkaline phosphatase -CIAP- 10u) has been added (45min at 37°C, in presence of protease inhibitors) prior to SDS-PAGE. Immunoblot analysis has been done using antibodies raised against TFIIE α and Ser5-P of Pol II. Signals were quantified (means \pm s.d.) and plotted in arbitrary units (au). The results are representative of three independent experiments.

panel d: streptavidin magnetic beads were incubated (when indicated, +) with biotinylated AdMLP, Pol II, TFIIA,-B,-D (TBP), rIIE $\alpha\beta$ /WT,-F and TFIIH/WT, in presence of whole cell extracts (WCE), ATP (200 μ M) and NTP (200 μ M). Immunoblot analysis (for TFIIE α and β , XPB, CyclinH, Ser5-P and SPT5) and the transcription activity (309nt run-off transcript) were assessed after 45min of incubation.

Reviewer #2

We thank reviewer#2 for finding that “there are some potentially important advances in this work”. According to her/his comments, new experiments have been performed, as outlined below, and several modifications have been done to improve the manuscript.

Fig. 1-

The results here are intriguing but they are not fully consistent across all of the assays. Consider panels D and E, and compare the effect of the E mutant on pol II levels (RPB1) at the promoter versus pol II levels at exon 4, particularly at 8 hr. At 8 hr, there is essentially no drops in gene expression or in pol II levels at the promoter because of the E mutant. However, there is far less pol II at exon 4 in the case of the mutant, which seems at odds with the gene expression result. The authors don't address this point. Importantly, these points are a hint that the E mutant has post-initiation effects, which is significant when considering some of the other results in this paper.

The lack of correlation between expression of the RAR β 2 gene and occupancy by RPB1 in a downstream region of the gene requires taking into account different parameters: (i) the transcriptional activation of the RAR β 2 gene is a cyclical process (see panel a, Annex Reviewer#2) (Le May et al., Mol Cell, 2010), the RAR β 2 mRNA having a relatively short half-life, (ii) the recruitment of Pol II has been analyzed at exon 4, which is located far away from the transcription start site, at ~140kb (iii) elongating Pol II (and factors such as SPT5) might be much more present at exon 4 between the selected times of t-RA treatment (6 and 8h).

An intimate correlation is not easy to establish between the cyclic profile of the RAR β 2 gene expression and the recruitment of transcription factors in a distal region. Extensive time-course analysis (at least every 30 min) should be undertaken to carefully follow the profile of recruitment of the general transcription factors all along the RAR β 2 gene (at +1, +1kb, +10kb etc... from the transcription start site). Nascent RNA should be analyzed in parallel for counting the nascent RAR β 2 RNA as a function of time. Here, our ChIP experiments was only to determine whether TFIIE mutations might lead to improper recruitments of transcription factors at the proximal promoter and what might be the consequences of a defective PIC formation in the ongoing transcription.

This point has been addressed in the manuscript (page 7): “The fact that signals for phosphorylated elongating Pol II have been observed at 8h while the RAR β 2 mRNA tend to decrease might be due to the cyclic profile of the RAR β 2 gene expression (Le May et al., Mol Cell, 2010) and the very distal position of exon 4 from the transcription start site (~140kb).”

Fig. 2-

The in vitro assay (as in panel E) is not well described- the reference is from 1991. Are assembly reactions still done for 15 min? If so, did the authors check to see if assembly was essentially complete in 15 min for both wt and mutant IIE?

In both abortive and run-off assays, preincubation was performed during 15min (at 25°C); this is now included in Material and Methods (see the “Transcription reactions” and “Abortive synthesis reactions” sections). These conditions were systematically controlled and were optimized a long time ago when we set up assays with recombinant proteins. In these conditions, the transcription machinery (including Pol II -RPB2-, TBP, TFIIF -RAP74-) is bound to the promoter (see in particular Fig. 3a), although the recruitment of TFIIE α and the CAK is disrupted in presence of TFIIE β mutation.

That is, is the mutant IIE defect related to failure in assembly, in function of the assembled complexes, or both?

According to our experiments, it seems that TFIIE mutations prevent the accurate formation of the pre-initiation complex, disturbing the recruitment and/or the positioning of IIE α and therefore of the CAK.

My impression is that such classic assays essentially report on a single round of transcription. This is important when comparing the effect of the E mutants, particularly D187Y, in the run-off versus the abortive assay. D187Y IIE works almost as well as wt in the abortive assay but much less well in run-off. Abortive synthesis of CAC is clearly multiround. Does this mean that complexes that contain D187Y IIE initiate well but fail to clear the promoter, perhaps because XPB cannot effectively support advance out to ~+10? Is a failure in transcription early, but post-initiation, somehow linked to the anomaly noted above with Fig. 1?

Our in vitro (abortive and run-off) assays showed that IIE/D187Y IIE as well as IIE/A150P worked much less than IIE/WT (Fig. 2). Furthermore, we repeatedly observed that transcription was more affected by IIE/A150P than /D187Y, which was not the case in our in vivo experiments (at least in our experimental conditions, Fig. 1b and 1c). However, subtle differences might be observed between the two TFIIE mutated cell lines during accurate kinetic analyses after t-RA treatment. Transcription failures might result from inaccurate recruitment and/or positioning within the PIC of mutated IIE. As rightly mentioned by the reviewer, we cannot exclude that complex containing IIE/D187Y might better initiate than IIE/A50P. Interestingly, (and in addition to the run-off and abortive transcription assays), the Pol II/Ser5 phosphorylation also seems to be less disrupted by IIE/D187Y (Fig. 6c). Accordingly, it is tempting to suggest that differences between D187Y and A150P might occur during either Pol II phosphorylation, DNA opening, first phosphodiester bond...

Fig. 3-

These results reinforce the well-established idea that IIE is important for the effective recruitment of IIH, and they advance our understanding by further discriminating between the individual roles of IIE alpha and beta. However, the functional points of several of the Fig. 3 results are not clear. For example, in panel A we see that without any IIE, the core of IIH is still recruited, at perhaps 40% of its optimal level. Do those complexes make any RNA (abortive or run-off)?

Neither transcript (new Fig2c, lane 1) nor trinucleotides synthesis (new Fig2e, lane 1) have been obtained in absence of TFIIE, which is in accordance with previous observations.

When either mutant IIE is used, the core of IIH is recruited as well as when wt IIE is used, but we know from Fig. 2 that these complexes are deficient to varying degrees in RNA synthesis assays. A potential problem which affects all experiments like those in this figure is the relationship between GTF assembly as assayed by western and the level of functional PICs assembled. Do the authors have any insight into this point- that is, what fraction of the apparent PICs actually make RNA? This is relevant in considering some of the other results. In particular, it will be shown that CAK is lost upon ATP addition and CAK retention can affect transcriptional competence of the complex. Is RNA synthesis the same (abortive and run-off) when complexes are assembled with core IIH only, as compared to complexes that have lost CAK as a result of ATP addition (and thus also have phosphorylated CTDs)?

Complexes assembled with Core-IIH did not promote efficient RNA synthesis, compared to complexes containing entire TFIIH (Coin et al., Mol Cell, 2006), suggesting that CAK is required for optimal transcription (see also below, Point Fig.5).

ATP pre-incubation disturbed the PIC, as observed by partial releasing of transcription machinery components, including TFIIE α/β , CAK and, to a lesser extent, Core-TFIIH (Fig. 5a, compare lanes 2 and 3). In these conditions, we observed a lower RNA synthesis when compared to that obtained without ATP pre-incubation (panel b, Annex reviewer#2).

Fig. 4-

Since the exact experimental procedure is not fully described, it is not clear if the assays with triptolide are meaningful. Since TPL is a covalent inhibitor, it takes many minutes for full inhibition of XPB to occur, while promoter opening is, as far as I know, very rapid. Thus, to conclude that promoter opening did not happen I would want to see preincubation with TPL for 10 min, only then followed by addition of ATP. Fortunately for the authors, I think they are on solid ground with the

XPB mutants, so the point about CAK loss prior to promoter opening seems well-supported. Without additional tests, I would not include the TPL results.

Triptolide has been systematically preincubated during 20min (at RT) with the DNA template and the general transcription factors (including TFIID) before addition of ATP. This experimental procedure has been detailed in the new version (Fig legend 4d).

Fig. 5-

Panel A indicates that a substantial fraction of IIE, XPB and pol II remain on the template after chase with all NTPs. Pol II retention can occur the end of template DNAs even after “run-off”, depending on the nature of the template end and the reaction conditions. However, based on the structure of the PIC and the initiating complex, it does not seem reasonable that any IIE or IIH should remain on the DNA after promoter clearance. This again raises the question of whether a substantial fraction, perhaps the majority, of the GTFs retained on the template are not in functional transcription complexes. Does this mean that most of the signal observed in the western comes from inactive complexes, when the complexes are assembled from purified components?

In a reconstituted in vitro transcription assay, we could never exclude the presence of inactive complexes resulting from non-correctly folded proteins and/or uncomplete PIC assembly. However, we wish to mention that, in our experimental conditions, almost all the Pol II was phosphorylated after ATP addition, which is illustrated by the complete conversion of RNA polymerase IIA to IIO (Fig. 5a, lanes 3 and 4). Such observation suggests that the majority of the polymerase bound to the DNA template might be integrated into complexes that allow it to become active.

As rightly mentioned, one cannot exclude that GTFs bound to the DNA template are not functional transcription complexes. A much longer period of incubation (beyond 45min) might increase the transcripts amount. However, in such conditions, previous works (performed to optimize in vitro transcription assays) showed that one could have degradation of DNA template and/or RNA transcript. In addition, a long time ago, elongating Pol II was shown to be tightly bound to the DNA template; very high salt concentrations (up to 1-1.5M Ammonium sulfate) failed to dissociate Pol II from DNA template. It is worthwhile to notice that we used a linear DNA template. Maybe additional factors (such as phosphatase or others) might be required to remove Pol II from this template. Moreover, the Pol II escape does not induce the removal of all the GTFs bound to the promoter; this also might require additional events.

This worry presumably dictated the panel C experiment with nuclear extract addition, whereas expected XPB and (mostly) IIE are lost on chase- but pol II is still retained. I think there is an important control missing from panel A. The complexes were incubated with ATP for 15 min, after which a 309 nt run-off RNA could be made. I found this surprising, since there is an extensive earlier literature showing that pol II PICs have a very limited lifespan after ATP incubation (see: JBC 271, p. 7245; EMBO J 16, p. 7468; JBC 287, p. 961). This suggests a clear question- if the complexes in Fig. 5 had been incubated for 15 min without ATP and then chased, how much run-off RNA would have been made? Is there a substantial loss of RNA synthesis upon extended incubation with ATP in this system?

We wish to mention to the reviewer that this experiment has been performed in triplicates and we always observed RNA synthesis upon ATP preincubation.

Since TFIIE can reactivate Pol II after ATP pre-incubation (Cabart et al., JBC, 2012), supplementations have been undertaken with increasing amounts of IIE α , following ATP treatment, washes and addition of NTPs (panel c, Annex reviewer#2). Supplementation of TFIIE α led to increase its presence at the promoter (lanes 5-6) and to increase RNA synthesis (panel d, lanes 4-5, Annex reviewer#2). These results strongly support previous observations (Cabart et al., JBC, 2012) and reinforce our idea that TFIIE is a key factor to stabilize PIC formation and its release contributes to the initiation.

Regardless of the answer, it is important to place the current work in the context of these well-established earlier findings. The THZ1 result (panel B, lanes 4 and 5) is very hard to explain. The

authors suggest that this may have resulted from residual CAK bound to the promoter, probably because of the synergy observed with added CAK and THZ1, but at least in Fig. 5 there is no evidence of any CAK in the complexes when additional CAK is not added (lanes 4, 5 of panel B). The authors interpret the Fig. 5B results to show that retention of CAK impedes RNA synthesis. One could imagine that CAK release is important in allowing XPB to effectively drive DNA into the PIC within the overall IIH complex, so blocking CAK release could inhibit open complex formation, consistent with CAK release before template opening. But, the add-back experiment is not a direct test of that idea. The bead-bound PICs had seen ATP and thus the templates were melted and the CTDs were modified before washing and re-addition of CAK. I am not sure what mechanism the authors have in mind for CAK inhibition in this case.

The CAK supplementation led to a significant increase of its recruitment, and a slight decrease of RNA synthesis (Fig. 5b, lanes 2-3).

The addition of THZ1 alone (lanes 4-5) led to a partial decrease of RNA synthesis (lanes 4-5), which might be due to the inhibition of residual CAK and to side effects (intercalating and/or interfering with DNA template) resulting from the concentrations of THZ1 used (1-10 μ M) and the absence of consistent amount of CDK7.

Addition of both CAK and THZ1 resulted in a higher (additive) transcription reduction (lanes 6-7). In this later case, THZ1-inactivated CAK was highly recruited at the DNA template, which might be related to the fact that TFIIE α (which has been added Fig. 5b) was maintained within the PIC (compare lane 1 and lanes 6-7). Strikingly, similar observations have been obtained upon addition of CAK/CDK7as (a CAK containing a CDK7 analog-sensitive mutant) (Larochelle et al., NS&MB, 2006; Mol cell, 2007) and 3-MB-PP1 (an ATP analog competitive inhibitor); the recruitment of CAK/CDK7as was much higher likely due to its anchoring at the promoter, TFIIE α being not released (new Supplemental Fig. 6a).

As mentioned by the reviewer, we cannot conclude that retention of CAK impedes RNA synthesis. However, supplementation of inactive CAK maintained the formation of a preinitiation complex containing an already phosphorylated Pol II, which prevented the subsequent steps of transcription. This part of the results has been modified (page 12) and supplemented Figure (Supplemental Fig. 6a) has been added.

“Addition of THZ1 led to a partial decrease of RNA synthesis (lanes 4-5), which may be due to the inhibition of residual CAK bound to the promoter, to side effects resulting from the concentrations of THZ1 used (1-10 μ M) and/or the absence of consistent amount of CDK7. Strikingly, addition of THZ1-inactivated CAK resulted in a higher transcription reduction (lanes 6-7). While such supplementation did not alter Pol II phosphorylation, a massive recruitment of THZ1-inactivated CAK has been observed, which might be related to the fact that TFIIE α was maintained within the PIC (compare lane 1 and lanes 6-7). Similar observations have been obtained upon addition of CAK/CDK7as (a CAK containing a CDK7 analog-sensitive mutant) (Larochelle et al., 2006; Larochelle et al., 2007) and 3-MB-PP1 (an ATP analog competitive inhibitor); the recruitment of CAK/CDK7as was much higher likely due to its anchoring at the promoter, TFIIE α being not released (Supplemental Fig. 6a). It is worthwhile to notice that supplementation with rIIE α /WT led to a slight increase of its recruitment at the promoter, without modifying the behavior of the CAK and the phosphorylation of Pol II (Supplementary Fig. 6b).”

Overall, there are a number of interesting findings in this paper but as I noted above, there are significant ambiguities in the results in several places. In particular, there are important points where considerably more could have been learned if changes in transcription complex assembly/composition had been linked to changes in ability to make RNA. Finally, I note an important typo- on p. 3 in the last paragraph, the list of the components of TFIH-Core does not include XPD.

We wish to remind the reviewer that XPD is not considered as a subunit of the Core subcomplex. Structurally, XPD is independent to the Core and CAK. These two subcomplexes are indeed bridged by XPD, which interacts with p44 and MAT1 of the Core or the CAK, respectively (Compe et al, NRMCB, 2012). Accordingly, we did not include XPD within the Core, XPD being presented in the following sentence.

Annex Reviewer#2

a

b

c

d

Legends – Annex reviewer#2

panel a: relative RAR β 2 gene expression have been measured by RT-PCR in WT (KI-IIE β /WT, dark boxes) and TFIIIE mutated cells (KI-IIE β /A150P, open boxes) after different times of t-RA treatment. The mRNA levels were normalized to the 18S RNA amount. The results are presented as n-fold induction relative to non-treated cells.

panel b: Streptavidin magnetic beads with biotinylated AdMLP, Pol II and the GTFs (TFIIA,-B,-D (TBP), rII α β /WT,-F and TFIIH/WT) were pre incubated 15min (as depicted in the scheme, left part) in presence (lane 2) or absence (lane 1) of ATP (200 μ M). The beads were washed and NTPs (200 μ M) were added. Transcription activity (309nt run-off transcript) were assessed after 45min of incubation. The results are representative of three independent experiments.

panel c: Streptavidin magnetic beads were pre incubated (when indicated, +) with biotinylated AdMLP, Pol II and the GTFs (TFIIA,-B,-D (TBP), rII α β /WT,-F and TFIIH/WT) in presence of ATP (200 μ M). After washes, NTP (200 μ M) and increasing amounts of rII α /WT were added. Immunoblot analysis (for TFII α , XPB, CyclinH, Ser5-P) was assessed after 45min of incubation. The results are representative of two independent experiments.

panel d: Streptavidin magnetic beads were pre incubated (when indicated, +) with biotinylated AdMLP, Pol II and the GTFs (TFIIA,-B,-D (TBP), rII α β /WT,-F and TFIIH/WT) in presence of ATP (200 μ M). After washes, NTP (200 μ M) and increasing amounts of rII α /WT were added. Transcription activity (309nt run-off transcript) was assessed after 45min of incubation. The results are representative of two independent experiments.

Reviewers' comments:

Reviewer #1 (Remarks to the Author):

This is a revised version of a manuscript I reviewed previously. The authors have, for the most part, responded to my concerns adequately and significantly strengthened the paper. There are some points that still need to be clarified, however, which I list below. These issues do not necessarily require re-review, but they are significant and should be addressed by the authors before acceptance and publication. My remaining, specific concerns are:

1. In response to my initial review, point 1, stating that a requirement for CDK7 in the exchange of TFIIE and DSIF had previously been demonstrated, and that this was important context for their own study, the authors added citations to two papers I suggested (refs. 53 and 54). But, as I stated in my initial review, these studies only reported confirmations of this function, which was first demonstrated in the 2012 NSMB paper by Larochelle et al. (now reference 55). This chronology is obscured by the way these papers are now cited (Larochelle et al. are still only cited for showing that TFIIE is a CDK7 substrate). This should be corrected (i.e., by citing the Larochelle paper either before or together with the Kelso and Nilson papers).
2. A more substantive issue concerns the "CAK add-back" experiments in Fig. 5b (and the new Supplemental Fig. 6a). Although the authors have modified the language and toned down their interpretations in response to my initial points 2 and 23, it is still very hard to figure out what is actually being shown by these results. The passage from p. 12 the authors quote in their rebuttal to point 23 illustrates this uncertainty; it is loaded with caveats and considerations of possible artifacts ("side effects"), making me wonder how much these results really add to the paper, and whether the study might be strengthened by their omission. Ultimately that is the authors' (and editor's) decision, but if these data are to be retained, I would ask clarification of one point: In the new blot strip added in the revision, what is the source of the increased TFIIE- α that shows up on the immobilized template when excess CDK7 and inhibitor are added (Fig. 6b, lanes 6, 7; Supp. Fig. 6a, lane 8)? As I understand the experimental setup, there should be no "free" TFIIE present during the transcription reaction (conducted after the beads were washed).
3. Conversely, the "Annex" included in the rebuttal, also in response to my point 23, shows a result (panel d) that actually strengthens the story, and should probably be included, i.e., that ATP addition is sufficient for TFIIE- α eviction but that all 4 NTPs (and therefore, possibly, transcription) are needed for removal of TFIIE- β and DSIF recruitment.
4. Although the authors removed the unsupported assertion that TFIIE- β or XPD mutations were affecting "positioning" of CAK from the Results section, this language, with its misleading promise of structural information, is retained in the Abstract (p. 2) and Introduction (p. 4, last paragraph).

Reviewer #2 (Remarks to the Author):

Review of NCOMMS-18-29809A

In my review of the original version of this manuscript, I acknowledged the importance of gaining a better understanding of the roles of the TFIIE subunits in PIC formation and in the initial stages of transcription. I had a number of concerns that I thought significant, but I was disappointed by the authors' responses.

For the Figure 2 results, I thought it was important to address the differences in abortive versus longer RNA synthesis in particular for the D187Y mutant and wildtype IIE, because this could be informative in understanding a role for IIE (probably through IIH) in supporting the earliest stages of transcription. The authors did provide comments on my point in their letter but I don't think those comments addressed the issue I raised, and in any event there were no changes in the revised mss.

I was particularly concerned with understanding the effect of ATP on the complexes as assayed in Fig. 3. Much earlier literature indicated that the activity of PICs is expected to be reduced by exposure to ATP alone. I was surprised that no control was provided to show to what extent that effect was observed in the complexes studied here. The authors did provide a control in the rebuttal letter- but that was not incorporated into the paper, and the question of ATP effects on complex activity was not discussed in the revised mss.

I was also concerned about the retention, or loss, of core IIE and IIH components when pol II advances to run-off as tested in Fig. 5. Panels b and c suggest very different results. At minimum this deserves some comment in the paper. Again, the authors discussed my points in their response letter but the mss has not changed.

In sum, I feel that many of the issues I raised in my original review have not been adequately addressed.

We would like to bring to your attention that previous questions of the reviewers gave us the impression of being engaged in a discussion with them. Accordingly, some points were not fully addressed in the former version of the manuscript. We wish to mention that the new version includes all modifications previously requested by the reviewers. Undoubtedly, their comments greatly improved the quality of the manuscript.

Reviewer #1 :

1. In response to my initial review, point 1, stating that a requirement for CDK7 in the exchange of TFIIE and DSIF had previously been demonstrated, and that this was important context for their own study, the authors added citations to two papers I suggested (refs. 53 and 54). But, as I stated in my initial review, these studies only reported confirmations of this function, which was first demonstrated in the 2012 NSMB paper by Larochelle et al. (now reference 55). This chronology is obscured by the way these papers are now cited (Larochelle et al. are still only cited for showing that TFIIE is a CDK7 substrate). This should be corrected (i.e., by citing the Larochelle paper either before or together with the Kelso and Nilson papers).

The Larochelle's paper has been cited together with the Kelso and Nilson papers (page 17).

2. A more substantive issue concerns the "CAK add-back" experiments in Fig. 5b (and the new Supplemental Fig. 6a). Although the authors have modified the language and toned down their interpretations in response to my initial points 2 and 23, it is still very hard to figure out what is actually being shown by these results. The passage from p. 12 the authors quote in their rebuttal to point 23 illustrates this uncertainty; it is loaded with caveats and considerations of possible artifacts ("side effects"), making me wonder how much

Ultimately that is the authors' (and editor's) decision, but if these data are to be retained, I would ask clarification of one point: In the new blot strip added in the revision, what is the source of the increased TFIIE- α that shows up on the immobilized template when excess CDK7 and inhibitor are added (Fig. 6b, lanes 6, 7; Supp. Fig.6a, lane 8)? As I understand the experimental setup, there should be no "free" TFIIE present during the transcription reaction (conducted after the beads were washed).

As suggested by the reviewer, since the experiments previously performed (Fig. 5b and Sup. Fig. 6a) did not seem to **"really add to the paper, and whether the study might be strengthened by their omission"**, we decided to remove these data from the manuscript.

3. Conversely, the "Annex" included in the rebuttal, also in response to my point 23, shows a result (panel d) that actually strengthens the story, and should probably be included, i.e., that ATP addition is sufficient for TFIIE- α eviction but that all 4 NTPs (and therefore, possibly, transcription) are needed for removal of TFIIE- β and DSIF recruitment.

As suggested by the referee, this figure **"that actually strengthens the story"**, has been added in the revised version (new Fig.5f) and the text has been modified, accordingly (page 12-13): "A partial removal of TFIIE β has been also observed in presence of ATP alone (Fig. 5f, lane 3). However, the elongation factor DSIF (SPT5) was not recruited when compared to what obtained in presence of NTP (lane 4), suggesting that ATP addition was sufficient for TFIIE α eviction but that all four NTPs were needed for complete removal of TFIIE β and DSIF recruitment."

4. Although the authors removed the unsupported assertion that TFIIE- β or XPD mutations were affecting "positioning" of CAK from the Results section, this language, with its misleading promise of structural information, is retained in the Abstract (p. 2) and Introduction (p. 4, last paragraph).

According to the reviewer's comment, "position" has been removed from the Abstract (page 2) and Introduction (page 4).

Reviewer #2 (Remarks to the Author): Review of NCOMMS-18-29809A

In my review of the original version of this manuscript, I acknowledged the importance of gaining a better understanding of the roles of the TFIIE subunits in PIC formation and in the initial stages of transcription. I had a number of concerns that I thought significant, but I was disappointed by the authors' responses.

For the Figure 2 results, I thought it was important to address the differences in abortive versus longer RNA synthesis in particular for the D187Y mutant and wildtype IIE, because this could be informative in understanding a role for IIE (probably through IIH) in supporting the earliest stages of transcription. The authors did provide comments on my point in their letter but I don't think those comments addressed the issue I raised, and in any event, there were no changes in the revised mss.

We wish to remind that it is relatively difficult to compare data obtained from abortive and run-off assays. While the abortive transcription assays have been performed during 30min with increasing amounts of recombinant IIEs, run-off transcription assays have been stopped after either 5, 10 or 20 min of incubation with similar amount of each recombinant TFIIE. Nonetheless, run-off and abortive transcription assays allowed us to observe that rIIE $\alpha\beta$ /A150P and /D187Y led to a lower RNA and CpApC synthesis (Fig.2c and 2e). Strikingly, and as rightly mentioned by the reviewer, differences can be observed between the two mutated forms of TFIIE. This point is now addressed in the manuscript (page 9): "The defect was more important with rIIE $\alpha\beta$ /A150P than /D187Y, which paralleled what was observed during in vitro run-off assays (Fig. 2c). Interestingly, the difference between the two mutated forms of TFIIE was more pronounced in run-off than in abortive assays; such observation suggested that early transcriptional steps might be differently affected by rIIE $\alpha\beta$ /A150P and rIIE $\alpha\beta$ /D187Y, the latter ensuring initiation but maybe failing subsequent steps."

I was particularly concerned with understanding the effect of ATP on the complexes as assayed in Fig.3. Much earlier literature indicated that the activity of PICs is expected to be reduced by exposure to ATP alone. I was surprised that no control was provided to show to what extent that effect was observed in the complexes studied here. The authors did provide a control in the rebuttal letter- but that was not incorporated into the paper, and the question of ATP effects on complex activity was not discussed in the revised mss.

Indeed, ATP addition results in modification during PIC formation, as previously observed Fig. 4. As suggested by the referee, this control experiment has been added in the manuscript (new Fig.5b). Furthermore, we added the experiments with TFIIE α supplementations, following ATP pre-incubation (new Fig. 5c and 5d). The text has been modified, accordingly (page 12):

"It is noteworthy that a lower RNA synthesis was obtained when an ATP pre-incubation was done (Fig. 5b), which was in agreement with previous observations showing that the activity of PICs was reduced by exposure to ATP alone (Dvir, 1996; Holstege, 1997). Since TFIIE can reactivate Pol II after ATP pre-incubation (Cabart, 2012), supplementations have been done with increasing amounts of IIE α , following ATP treatment, washes and addition of NTPs (Fig. 5c). Such supplementation led to increase the presence of TFIIE α at the promoter (lanes 5-6) and to improve RNA synthesis (Fig. 5d, lanes 4-5), as previously observed (Cabart, 2012)."

I was also concerned about the retention, or loss, of core IIE and IIH components when pol II advances to run-off as tested in Fig. 5. Panels b and c suggest very different results. At minimum this deserves some comment in the paper. Again, the authors discussed my points in their response letter but the mss has not changed. In sum, I feel that many of the issues I raised in my original review have not been adequately addressed.

This point has been discussed in the new version (page 15-16):

"The PIC formation assays questioned about the retention, and loss, of transcriptional components while Pol II advances to transcribe and is retained on the DNA template (Fig. 5a, lane 4). In a reconstituted in vitro transcription assay, we cannot exclude Pol II retention at the end of DNA template, a situation requiring additional factors for the release of the polymerase. Furthermore, formation of inactive complexes can occur, resulting from non-correctly folded proteins and/or

uncomplete PIC assembly. However, in our experimental conditions, almost all the Pol II was phosphorylated after ATP addition, which is illustrated by the complete conversion of RNA polymerase IIA to IIO (Fig. 5a, lane 3). Such observation suggests that the majority of the polymerase bound to the DNA template might be integrated into complexes that allow it to become active. Interestingly, a selective (i.e. TFIIE and TFIIH) but not a complete removal of general transcription factors has been observed once Pol II is engaged in elongation (lane 4), as previously suggested (Zawel, 1995). Such dynamic recruitment/release of factors illustrate the complexity of the sequential events that are required to accomplish promoter clearance (Luse, 2013).“

REVIEWERS' COMMENTS:

Reviewer #1 (Remarks to the Author):

The authors have now addressed my remaining concerns; I recommend publication in the present form.

Reviewer #2 (Remarks to the Author):

Review of NCOMMS-18-29809B

I was pleased to see that the authors have substantially addressed my concerns from my earlier review, particularly with regard to including controls about the effect of ATP on complexes. They had performed these controls before but previously they were only available to the reviewers.